# Characterisation of a synthetic Archeal membrane reveals a possible new adaptation route to extreme conditions

Marta Salvador-Castell[1], Maksym Golub[2,3], Nelli Erwin[4], Bruno Demé[3], Nicholas J. Brooks[5], Roland Winter [4], Judith Peters [2,3 ✉] & Philippe M. Oger [1 ✉]

It has been proposed that adaptation to high temperature involved the synthesis of monolayer-forming ether phospholipids. Recently, a novel membrane architecture was proposed to explain the membrane stability in polyextremophiles unable to synthesize such lipids, in which apolar polyisoprenoids populate the bilayer midplane and modify its physico-chemistry, extending its stability domain. Here, we have studied the effect of the apolar polyisoprenoid squalane on a model membrane analogue using neutron diffraction, SAXS and fluorescence spectroscopy. We show that squalane resides inside the bilayer midplane, extends its stability domain, reduces its permeability to protons but increases that of water, and induces a negative curvature in the membrane, allowing the transition to novel non-lamellar phases. This membrane architecture can be transposed to early membranes and could help explain their emergence and temperature tolerance if life originated near hydrothermal vents. Transposed to the archaeal bilayer, this membrane architecture could explain the tolerance to high temperature in hyperthermophiles which grow at temperatures over 100 °C while having a membrane bilayer. The induction of a negative curvature to the membrane could also facilitate crucial cell functions that require high bending membranes.

[1] Univ Lyon, INSA Lyon, CNRS UMR5240, Villeurbanne, France. [2] Université Grenoble Alpes, CNRS, LiPhy, Grenoble, France. [3] Institut Laue Langevin, Grenoble, France. [4] Faculty of Chemistry and Chemical Biology, Technische Universität Dortmund, Dortmund, Germany. [5] Department of Chemistry, Imperial College London, London, England. ✉email: peters@ill.eu; philippe.oger@insa-lyon.fr

Cell membranes are two-dimensional solutions composed of proteins and lipids, such as glycolipids and phospholipids[1–3]. However, the plasma membrane is not a mere barrier which surrounds the cell content to protect it. Cell membranes are the energetic factory of the cell and are indispensable for numerous cellular processes. Therefore, it is crucial for cells to maintain this barrier in a functional state, which is a state referred to as the liquid–crystalline phase, regardless of environmental conditions. Many physico-chemical parameters can disrupt the structure and function of membranes and consequently alter the plethora of cellular functions that depend on their integrity. High temperatures, for example, result in higher molecule agitation to the point that ordinary membrane lipids may pack too loosely to maintain the functional selectively permeable liquid-crystalline structure. As a response to varying environmental conditions, cells adapt their phospholipid membrane compositions to keep their membrane functional (i.e. homeoviscous adaptation). Over time, membrane adaptation has translated into several adaptation routes in the membranes of the most extremophilic organisms, e.g. Archaea. Indeed, regardless of the environmental constraint (pH, temperature, salinity, etc.), Archaea are usually found to be adapted to the most extreme values of these parameters. This ability has been linked to their specific membrane lipids, which differ from those found in Eukarya and Bacteria: they are isoprenoid hydrocarbon chains linked by ether bonds to sn-glycerol-1-phosphates (Fig. 1a)[4,5]. Ether-based polyisoprenoid lipids form more compact, more impermeable membranes, which may explain their tolerance to the extremes. Furthermore, Archaea may also synthesize bi-polar phospholipids (tetra-ethers)[6]. These membrane-spanning lipids self-assemble into monolayers that are highly stable due to restricted motility of the hydrocarbon chains and the presence of the isoprenoid methyl groups that block water penetration through the membrane[7,8]. Thus, the lipid monolayer tightly compacts the cell membrane[9], reduces its fluidity and permeability[10] and maintains it in the range of values of the functional state[11]. It is well documented that Archaea adapted to live under extreme temperatures accumulate bipolar ether-linked isoprenoid lipids in their membranes[6,12,13]. The discovery of monolayer forming, ether lipids in the membranes of the most hyperthermophilic bacteria gave further support to the view that tetraether lipids were the adaptation route to life at high temperatures[14]. In these Archaea, the homeoviscous adaptation involves the regulation of the number of cyclopentane rings, crosslinked chains and the ratio of di- and tetraether lipids, in order to maintain the membrane in its functional state. Recent observations however have challenged this dogma. First, a tetraether-containing membrane might not be a prerequisite for heat tolerance, as evidenced by the absence of tetraether lipids in the membranes of *Aeropyrum pernix* ($T_{max} = 100\,°C$, $T_{opt} = 90–95\,°C$)[15] or *Methanopyrus kandleri*[16], the alleged current high temperature record holder with a maximal growth temperature for growth of 122 °C at 20 MPa ($T_{opt} = 100\,°C$ at atmospheric pressure)[17]. Second, numerous hyperthermophilic archaea produce a mixture of tetra- and diether lipids, amongst which tetraether lipids might represent <10% of membrane lipids[18–20]. This finding raises questions about the thermal stability of the archaeal lipid bilayer but also suggests that archaea may have developed adaptive routes to hyperthermophily that might not require tetraether lipids.

The key molecular and assembly interactions that allow hyperthermophiles with membrane bilayers to survive under high-stress conditions remain a hotly debated topic. A recent membrane architecture model[21] (Fig. 1b) suggests that apolar polyisoprenoid molecules may act as structural membrane components, increasing the stability of the membrane at higher hydrostatic pressures and temperatures. This model predicts that apolar isoprenoids would populate the midplane of the bilayer, thereby altering its physico-chemical properties while causing significant changes to lipid dynamics in the membrane[22]. The presence of the intercalant would limit charge transfer between the two membrane leaflets, decreasing proton and water permeability and increasing membrane rigidity[22], thus, extending the conditions for stability and functionality, and thereby providing a rationale for the ability of these hyperthermophiles to withstand temperatures above the boiling point of water as proposed[21]. To date, the insertion of alkanes in a membrane composed of bacterial/eukaryal type phospholipids has been demonstrated[23] and proven to strongly, and negatively, impact membrane structure and functionality[24]. In contrast, in archaeal bilayer, the insertion is proposed to extend the stability domain and enhance the functionality of the membrane[21,22]. If confirmed, this hypothesis has numerous implications: amongst them, it would provide evidence for a novel putative adaptation route of the membrane to high temperature in Archaea, which could be applicable to Archaea ancestors predating the invention of bipolar lipids and which would be congruent with a possible origin of life in deep sea hydrothermal vent systems[25,26]. This architecture could also

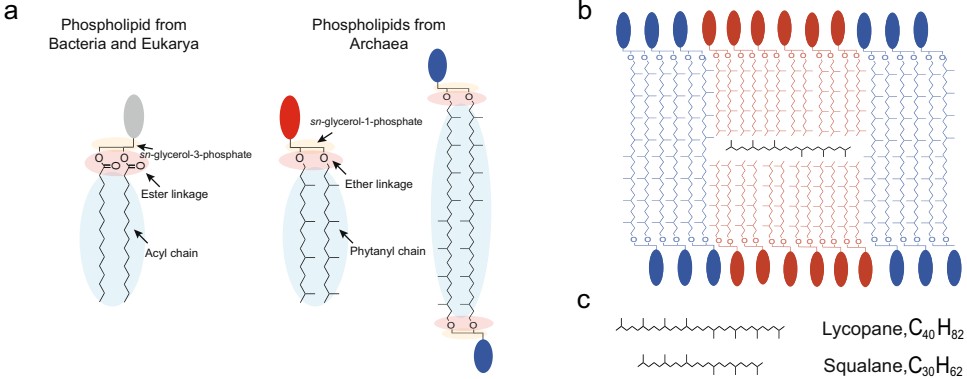

**Fig. 1 The lipid divide and membrane organization in *Thermococcus barophilus*. a** Skeletal representation of phospholipids from Bacteria and Eukarya (left) and from Archaea (right). The arrows indicate the main differences between them. **b** The proposed membrane ultrastructure for the polyextremophilic archaeon *Thermococcus barophilus*. The membrane is composed of ca. half of monolayer-forming tetraether phospholipids (in blue with a blue polar headgroup) and half of bilayer-forming diether phospholipids (in red with a red polar headgroup). It is organized in membrane domains in monolayer or bilayer. The midplane of the bilayer is populated by **c**) polyunsaturated polyisoprenoids of 30 (squalane-derivatives, minor fraction) to 40 carbons (lycopane-derivatives, major fraction, black molecule).

imply the existence of membrane domains of divergent compositions in the archaeal membrane, and hence of divergent physico-chemical properties, and the ability of membrane-lateral structural functionalization in Archaea, along with all the known consequences of this possibility on membrane-assisted cellular processes[27]. This feature may possibly predate the separation of the two prokaryotic domains[20].

Here, we present an extensive experimental study of a simple synthetic membrane model, mimicking an archaeal membrane, demonstrating that the apolar polyisoprenoid squalane can insert at the midplane of the bilayer and that it strongly modifies membrane properties, providing strong support for the possibility that apolar intercalants may constitute one of the evolutionary routes to high-temperature and high-pressure adaptation in prokaryotes from early to contemporary cells.

## Results and discussion

**Apolar isoprenoids intercalate in the midplane of an archaeal lipid bilayer.** Haines[22] and Cario and colleagues[21] have proposed that the location and molecular orientation of apolar intercalants with respect to the membrane plane is a key parameter in the novel cell membrane model proposed for polyextremophilic archaea, since it is supposed to control the membrane's physico-chemical properties and the extent of its stability. To locate the intercalant precisely, we have constructed simplified synthetic archaeal membrane analogs to allow the precise control of their composition, and we have taken advantage of contrast variation neutron scattering, which can be used to selectively highlight different areas within the structure (see Methods section) and

provide a detailed map of the membrane's assembly structure and molecular interactions.

Synthetic archaeal membrane bilayer analogs were produced from a 9:1 molar ratio of two synthetic archaeal-like lipids, 1,2-di-O-phytanyl-*sn*-glycero-3-phosphocholine (DoPhPC) and 1,2-di-O-phytanyl-*sn*-glycero-3-phosphoethanolamine (DoPhPE), which reproduces the bilayer structure of the *Thermococcus barophilus* membrane (Fig. 1b, red lipids/Fig. 2a, d). The rationale for the choice of synthetic vs. natural lipids or membranes is several-fold. First, due to the low growth yield of the Archaea potentially harboring this membrane ultrastructure, e.g. *M. kandleri* or *T. barophilus*, it was not possible to obtain pure phospholipids from these species in quantities sufficient for the different experiments and biological repeats. Second, membrane lipid compositions in natural membranes are very finely tuned, which may generate important composition variations in the different batches of purified natural lipids. Furthermore, these variations may stay unnoticed since not all polar headgroups are accessible to analysis[28], but may still impact membrane parameters. Third, in contrast, using synthetic lipids allows for a precise modelization of the data, and for a precise allocation of the molecular interactions, better describing the behavior of the architecture as a function of the parameters tested.

One of the constraint of this approach is that there are no archaeal lipids available commercially to date, and no phospholipids harboring all the archaeal lipids features, especially the *sn*-glycerol-1 phosphate, e.g., the archaeal stereochemistry of the glycerol moiety. Thus, we have worked with synthetic analogs which have a glycerol moiety in the bacterial stereochemistry instead of the archaeal. Studies have always considered this

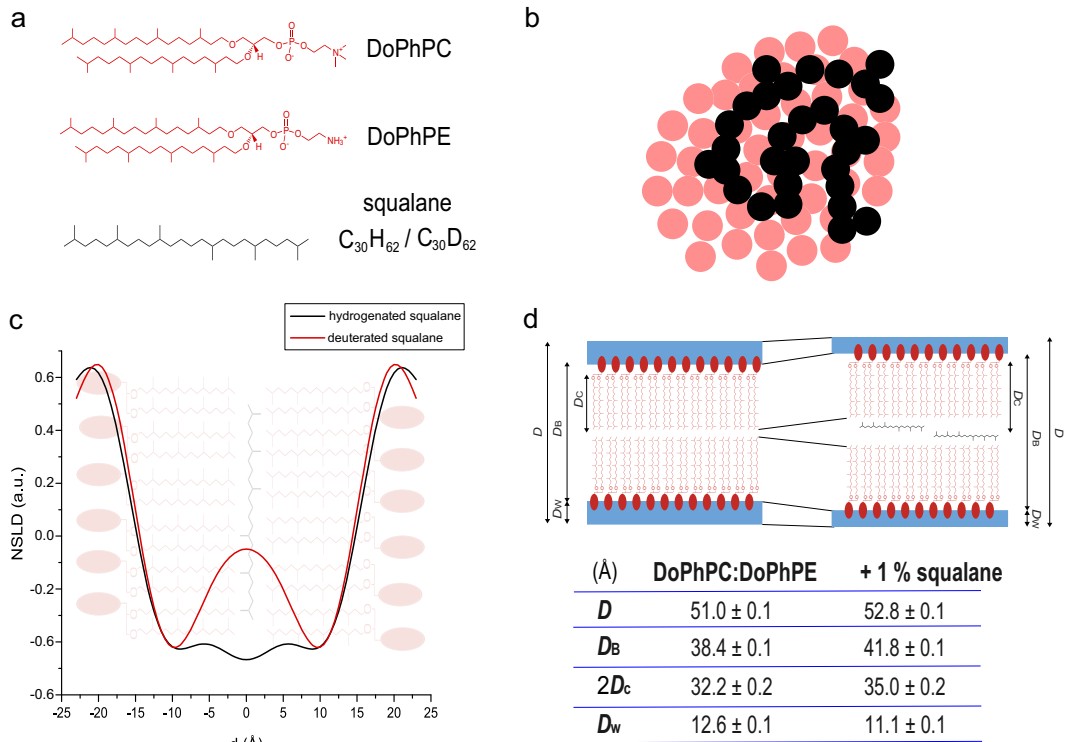

| (Å) | DoPhPC:DoPhPE | + 1 % squalane |
|---|---|---|
| $D$ | 51.0 ± 0.1 | 52.8 ± 0.1 |
| $D_B$ | 38.4 ± 0.1 | 41.8 ± 0.1 |
| $2D_C$ | 32.2 ± 0.2 | 35.0 ± 0.2 |
| $D_W$ | 12.6 ± 0.1 | 11.1 ± 0.1 |

**Fig. 2 Apolar isoprenoid molecules as structural membrane components placed in the midplane bilayer. a** Skeletal formula of the lipids used: 1,2-di-O-phytanyl-*sn*-glycero-3-phosphocholine (DoPhPC), 1,2-di-O-phytanyl-*sn*-glycero-3-phosphoethanolamine (DoPhPE), and 2,6,10,15,19,23-hexamethyltetracosane (squalane), hydrogenated, and deuterated. **b** Sketch of a top view of the interaction of the apolar lipid squalane (methyl groups represented as black spheres) in the midplane of the reconstructed archaeal-like lipid bilayer membrane (lipid isoprenoid chain extremities are represented in red). **c** Neutron scattering length density (NSLD) of DoPhPC:DoPhPE (9:1) + 1 mol% hydrogenated squalane (black) and deuterated squalane (red) measured on D166 (ILL, Grenoble, France). **d** Structural parameters: lamellar repeat spacing (D), bilayer thickness ($D_B$), hydrocarbon region thickness ($2D_C$), and water layer thickness ($D_W$) for bilayers of DoPhPC:DoPhPE (9:1) in the absence and presence of 1 mol% squalane.

difference in stereochemistry should not be expected to influence the physico-chemical behavior of the synthetic membrane. To date, the nature of the polar headgroups of *T. barophilus* or *M. kandleri* lipids remains largely unknown. The choice of the synthetic lipids limited the current work to phosphocholine and phosphoethanolamine polar headgroups. Fortunately, these phospholipids have been reported frequently in Archaea[18], and therefore should be representative to some extent of the behavior of natural archaeal lipids. Finally, we have chosen squalane (six isoprenoid groups, $C_{30}H_{62}$) as a model of apolar polyisoprenoid since Archaea synthesize apolar polyisoprenoids composed of four to eight isoprenoids groups (20–40 carbons). *T. barophilus* synthesizes in majority lycopane[21], a 40 carbon polyisoprenoid (Fig. 1c) and a 35 and 30 carbon polyisoprenoid to a lesser extent. Squalane is the synthetic isoprenoid homolog of the C30 polyisoprenoid synthesized by *T. barophilus*. It has been introduced here at a concentration of 1 mol% corresponding to the polyisoprenoid concentration found in *T. barophilus*[21] and many other extremophiles[29].

Protonated (H-squalane) and perdeuterated (D-squalane) pure squalane were introduced into the membrane at 1 mol% (Fig. 2, black molecule). Neutron diffraction data[30] was acquired at the D16[31] diffractometer of the Institut Laue-Langevin (ILL) in Grenoble, France, on oriented stacks of reconstructed synthetic membranes deposited on ultraclean silicon wafers[32]. The neutron scattering length density (NSLD) at different squalane H-D contrasts (Fig. 2c) shows the expected membrane profile for H-squalane-containing membranes with a maximum scattering length density at the lipid phosphate groups and a minimum at the methyl groups. D-squalane-containing membranes show a marked increase in NSLD at the center of the membrane, clearly locating the squalane in the midplane of the archaeal lipid bilayer. The width of the D-squalane peak in the NSLD is consistent with the squalane molecules being predominantly oriented parallel to the membrane surface and perpendicular to the polar lipid isoprenoid chains.

The NSLD data offer a range of additional structural parameters for the archaeal phospholipid bilayer[33], which include the thickness of the interlamellar water layer ($D_W$), of the hydrophobic core ($D_C$), the mean thickness of the membrane ($D_B$), and the lamellar repeat distance (d-spacing, $D$) (see panel d of Fig. 2 for details on the precise boundaries for each parameter). The d-spacing represents the periodicity of the bilayer structure, e.g., the distance between two bilayers in the stacks, including the interlamellar water thickness. In the presence of 1 mol% squalane, the d-spacing increases from 51.0 Å to 52.8 Å (Table in Fig. 2d). The Gibbs-Luzzati bilayer thickness ($D_B$), which corresponds to the distance between the center of the polar headgroups in the two membrane leaflets, is equal to 38.4 Å in the absence of squalane, which is similar to that found previously by molecular dynamics simulations[9]. In the presence of 1 mol% squalane, $D_B$ increases to 41.8 Å. Similarly, the thickness of the hydrocarbon chain region ($2D_c$) increases from 32.2 Å in the absence to 35.0 Å in the presence of 1 mol% squalane. All these parameters are consistent with the apolar molecule being inserted into the bilayer midplane, and increasing the thickness of the hydrophobic region of the membrane. At the same time, we observed a decrease in the water thickness between bilayers ($D_w$) from 12.6 Å to 11.1 Å in the presence of squalane (Table in Fig. 2d).

**Squalane induces the formation of non-lamellar phases in archaeal lipid membranes**. Low concentrations of apolar molecules can substantially influence the assembly of membranes, for example, disrupting membrane ultrastructure as in alkane-induced anesthesia[24] or inducing phase transitions between

lamellar and non-lamellar cubic or hexagonal phases[34]. Such structural transitions can be driven by the apolar alkanes' ability to partition between the bilayer leaflets and release packing frustration that would otherwise energetically prohibit the formation of curved phases[35]. Lamellar, cubic and inverted hexagonal phases possess different intrinsic curvatures[36] that have an impact on the energy necessary to bend the membrane[37]. Changes in membrane curvature are critical to a wide range of cellular processes, including cell membrane fusion/fission and lipid sorting, which require substantial membrane rearrangements[38–40].

Lipids spontaneously organize in different phases, the most well-known being the lamellar phase, e.g., the lipid phase of the plasma membrane. But depending on the local environment, they can also organize into so called non-lamellar phases, such as the cubic or hexagonal phase (see left top panel in Fig. 3). We used small-angle X-ray scattering (SAXS) to explore the possibility of squalane-induced phase transitions by determining the structure of fully hydrated synthetic archaeal-like lipids (DoPhPC:DoPhPE (9:1)) and the impact of 1 mol% squalane on their self-assembled structure as a function of temperature (10 to 85 °C, 15 °C steps) and hydrostatic pressure (1 to 1000 bar, 50 bar steps).

Our results reveal that in this membrane model system, non-lamellar phases (inverted bicontinuous cubic and inverted hexagonal) are only formed in presence of squalane, and that these phases are also promoted by high temperatures. In the absence of the apolar intercalant, only lamellar phases were found regardless of the temperature (T) and pressure (P) (Table in Fig. 3), as can be seen in the ratios (1:2:3…) of the scattering vectors at which Bragg peaks are observed. There was no clear gel-to-liquid or liquid-to-gel phase transition. This absence of a clear phase transition is a characteristic of archaeal diether lipid bilayers which is a major difference with fatty acid based-phospholipids of bacteria and eukarya[18,41]; however, the coexistence of two lamellar phases can be observed (Fig. 3 right top) up to at least 70 °C. The small difference observed in their d-spacing suggests that they are quite similar in structure. Several reasons could lead to this, such as a variation of the lipid composition due to lipid partitioning affecting the polar or apolar lipids, or both, or a tilting of the lipids in the membrane. The data acquired here does not provide any additional information about the distinction between these coexisting lamellar phases. At the highest temperatures tested, the scattering pattern became rather broad, indicating a loss of lamellar ordering, which is consistent with the known stability range for this type of lipid bilayer. Increasing hydrostatic pressure had a very limited impact on the bilayer structure (Table in Fig. 3 bottom). With 1 mol% squalane, two types of non-lamellar phases were observed (Fig. 3 right). Below 55 °C, coexistence between a cubic and a lamellar phase was induced, whereas above 55 °C, the lamellar phase transformed into an inverse hexagonal phase that coexisted with the cubic phase up to at least 85 °C. Again, no phase transition was observed as a function of hydrostatic pressure, but the lattice parameters of all phases increased with increasing pressure, which is consistent with the ordering of the lipid hydrocarbon chains (Supplementary Fig. 1). A number of factors are known to promote the formation of non-lamellar phases in phospholipid bilayers: headgroup dehydration, interaction with multivalent ions, acyl chain unsaturation, and increases in chain length[42,43]. These factors are unlikely to be significant in our membrane system, since the polar headgroups are fully hydrated and the acyl chains are not varied. Thus, squalane must be the factor promoting phase transitions to the cubic and hexagonal phase, most likely by allowing structural flexibility at the bilayer midplane and negative curvature of the lipid interface. Interestingly, a similar squalane-induced lamellar-to-hexagonal phase

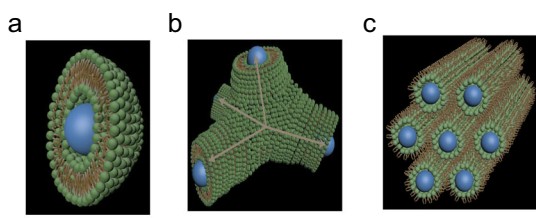

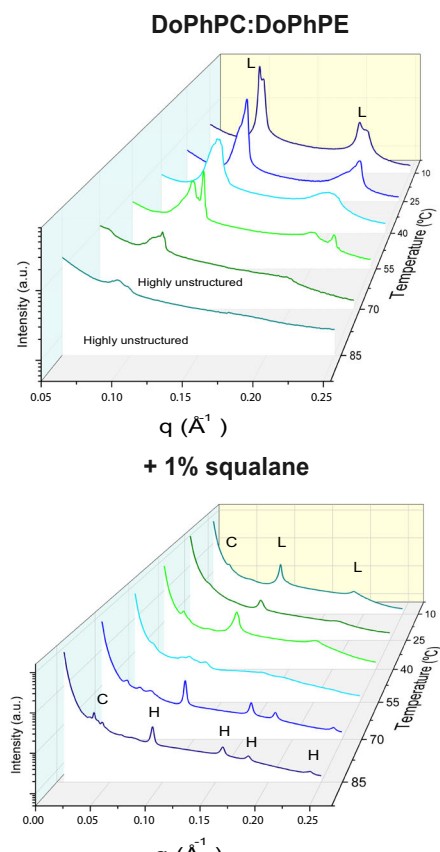

**DoPhPC:DoPhPE**

**+ 1% squalane**

| (°C) | Lipid phase | | Lipid phase | |
|---|---|---|---|---|
| | d-spacing | | d-spacing | |
| | 1 bar | 1000 bar | 1 bar | 1000 bar |
| **10** | Lamellar | | Lamellar | |
| | 62.7 Å ± 0.6 | 62.9 Å ± 0.6 | 67.6 Å ± 0.3 | 69.8 Å ± 0.3 |
| | | | Cubic | |
| **25** | Lamellar | | Lamellar | |
| | 60.0 Å ± 0.6 | 60.2 Å ± 0.6 | 69.6 Å ± 0.7 | 72.2 Å ± 0.7 |
| | | | Cubic | |
| **40** | Lamellar | | Lamellar | |
| | 64.6 Å ± 0.9 | 64.9 Å ± 0.9 | 69.6 Å ± 0.5 | 72.2 Å ± 0.5 |
| | | | Cubic | |
| **55** | Lamellar | | Hexagonal | |
| | 63 Å ± 1 | 63 Å ± 1 | 88 Å ± 1 | 93 Å ± 1 |
| | | | Cubic | |
| **70** | Lamellar | | Hexagonal | |
| | N.D. | N.D. | 82.0 Å ± 0.3 | 90.0 Å ± 0.3 |
| | | | Cubic | |
| **85** | Lamellar | | Hexagonal | |
| | N.D. | N.D. | 80.1 Å ± 0.3 | 84.8 Å ± 0.3 |
| | | | Cubic | |

**Fig. 3 Squalane induces lipid self-assembly in highly curved non-lamellar phases.** Top left: 3D representations of different lipid phases: **a** lamellar, **b** Pn3m (cubic), and **c** hexagonal. Table: lipid phases present in DoPhPC:DoPhPE (9:1) in the absence or presence of 1 mol% squalane at different temperatures and hydrostatic pressures. The lattice parameters (d-spacing) obtained at 1 bar and 1000 bar for the lamellar and hexagonal phases measured at I22 (DLS, Didcot, UK) are given. N.D. stands for "Not Detected". Right: radial SAXS intensities of DoPhPC:DoPhPE (9:1) in the absence (top) and presence of 1 mol% squalane (bottom) at different temperatures. Phases could be identified by the ratios of the scattering vectors at which Bragg peaks are located (lamellar (L) phase: 1:2:3; cubic (C) phase: $\sqrt{2}:\sqrt{3}:\sqrt{6}$; and hexagonal (H) phase: $1:\sqrt{3}:\sqrt{7:3}$).

transition has also been reported in a bacterial lipid system, although the precise location of squalane was not identified in that case[34].

Dynamic modulation of membrane bending is crucial to the function of cells, especially in membrane fusion, which is critical to membrane trafficking. The essential step in non-leaky membrane fusion is the rearrangement of the lipid molecules from the two apposed membranes to form a single continuous membrane. During this process, transitory isotropic structures are formed, which resemble the local structure of cubic phases[44,45]. Specific membrane curvatures yield membrane domains with specialized roles in membrane folding, protein incorporation, and enzymatic activities[43,46–48] (Fig. 4). These precise curvatures usually occur through the synthesis of at least one non-lamellar structure-forming component, which is often the synthesis of large-negative-curvature lipids (e.g., with phosphatidylethanolamine, phosphatidylserine, or phosphoric acid as polar head groups) and large-positive-curvature lipids (e.g., with phosphatidylinositol as polar head group)[49]. The presence of non-zero-curvature lipids is essential to decrease the otherwise large bending energy driven by protein interactions. In microorganisms, such as *Escherichia coli* or *Acholeplasma laidlawii*, that maintain a constant level of non-lamellar structure-forming lipids in the liquid-crystalline phase, this leads to an increase in the lipid conformational dynamics of the system[50]. The results we have obtained in this study on an archaeal membrane analog suggest that molecules such as squalane or other apolar hydrocarbons

inserted at the midplane in the archaeal bilayer membranes might play a similar role to allow the release of lipid chain frustration in the membrane and thus have a similar role of membrane regulators as seen for sterols derivatives in bacteria and eukarya[51]. Supporting this view, squalene, an unsaturated derivative of squalane, was found to represent ca. 20% of the lipid content of the purple membrane of the hyperthermophilic archaeon, *Halobacterium salinarum*. The purple membrane is ca. 75% percent protein by weight, formed by bacteriorhodopsin trimers that pack in a hexagonal lattice. To accommodate this high density of protein requires a membrane with, at least locally, numerous domains of high negative and positive curvature. In this context, squalene could be filling the voids left in the highly curved membrane and help reduce chain frustration. Further experiments would be needed to confirm the precise location of squalene inside the purple membrane.

**Squalane increases water permeability and reduces proton permeability under extreme conditions.** Biological membranes act as barriers to solute diffusion and play a central role in energy storage and processing via ion gradients[52]. The permeation of nonionic solutes is directly related to membrane fluidity and is affected by different factors, for example, the acyl chain length or the level of unsaturation in the phospholipid hydrocarbon chains[53]. If the area between the two membrane leaflets is populated, then apolar organics will also be expected to impact

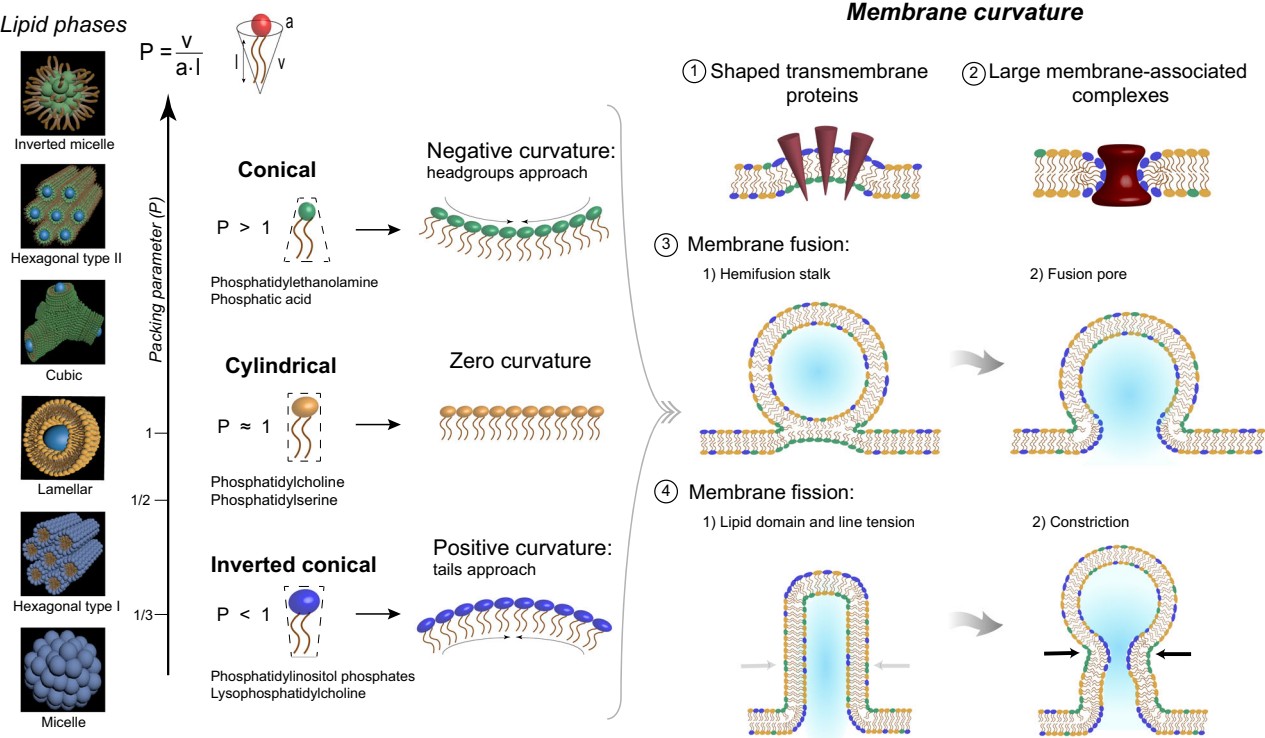

**Fig. 4 Sketch of different lipid phases that can be detected by SAXS and their significance for cell membranes.** Membrane curvature and relationship with lipid phases detected by SAXS. Left, different lipid phases detected by SAXS and ordered by their packing parameter ($P_p = v/a \times l$, where $v$ is the molecular volume, $l$ is the molecular length, and $a$ is the molecular area at the hydrocarbon-water interface). Molecules with high packing parameter (i.e., small head groups) have a conical geometry and tend to bend to negative curvatures (green), contrary to molecules with little $P$, which have an inverted conical geometry and bend to positive curvatures (blue). Right, clusters of conical or inverted conical lipids indicate the different curvatures of the membrane: negative (green) and positive curvatures (blue). 1–2: Different proteins anchor to membrane that require characteristic curvatures to be functional. 3: Process of membrane fusion, figures represent the two main steps where non-zero curvatures are needed. 4: Process of membrane fission, this process also needs characteristic membrane curvatures.

solute permeation. Three molecular models have been proposed for proton leakage across lipid bilayers: the "defect" mechanism[54], the "water wire" mechanism[55] and the "water cluster" model[22]. Despite differences in proton transport mechanisms, all three models suggest that any hydrocarbon in the center of the lipid bilayer will serve as an inhibitor of proton leakage. Thus, to validate the model proposed for hyperthermophilic archaea, we measured the membrane fluidity, solute and proton permeation across the synthetic archaeal membrane analog as a function of squalane concentration.

Membrane fluidity was assayed as a function of P and T using laurdan as a reporter probe[56,57] of the membrane fluidity of unilamellar lipid vesicles. Laurdan is a fluorescent probe for which emission wavelength is determined by the water content of the lipid bilayer's headgroup region, i.e., is sensitive to the phase state of the membrane[56]. The laurdan generalized polarization (GP) reports on the fluidity of the membrane. When the probe is placed at a low polarity or high polarity environment, it presents a blue or green emission, respectively. In membrane composed of ester-linked phospholipids, laurdan GP values of approximately 0.5 denote ordered gel phases, whereas values below zero are typical for fluid-like lipid phases. Due to the absence of a carboxyl group in ether bound lipids, such as those used in the current work, the position of laurdan in the membrane is somewhat different, and the impact on laurdan GP is still debated[58]. The insertion of laurdan may also be impacted by the presence of the polyisoprenoid chains instead of acyl chains. However, regardless these differences, a decrease in GP will always be associated with an increase of polarity of the laurdan environment. First, the

temperature-dependent laurdan GP values measured for membranes in the absence and presence of 1 mol% squalane appeared similar (Fig. 5a–c). Up to approximately 45 °C, we observed a decrease in the laurdan GP, indicating increased conformational disorder in the lipid system as would be expected upon increasing the temperature. Thereafter, the laurdan GP value increased again, indicating the slow and gradual replacement between two fluid phases of very similar parameter values, with the second phase displaying a higher laurdan GP. We observed no sharp phase transition in contrast to what would be expected for bacterial/eukaryal phospholipids. This behavior is quite characteristic of all archaeal, bilayer forming, lipid systems studied to date, for which no marked phase transitions have been observed in the 0–100 °C temperature range[41]. In contrast, small but visible phase transitions are evidenced in monolayer forming, tetraether archaeal lipid systems[7]. The laurdan fluorescence data are consistent with the coexistence of fluid phases and the effect of temperature on the phases observed by SAXS (Fig. 3 right). One should note that the slow transition observed here between two lamellar phases does not correspond to the lamellar to non-lamellar phase transition observed by SAXS at the same temperature. Indeed, the experimental conditions are different for both experiments. SAXS is performed at high concentrations of lipids (5 mg in 20 mg of water), which essentially yields water saturated lipid powders. This high concentration of lipids facilitates the interactions between bilayers, reduces the frustration of the lipids, and therefore enables the formation of non-lamellar phases. In contrast, vesicles are formed from lipid suspensions of much lower concentration (3 mg in 1 ml) in order

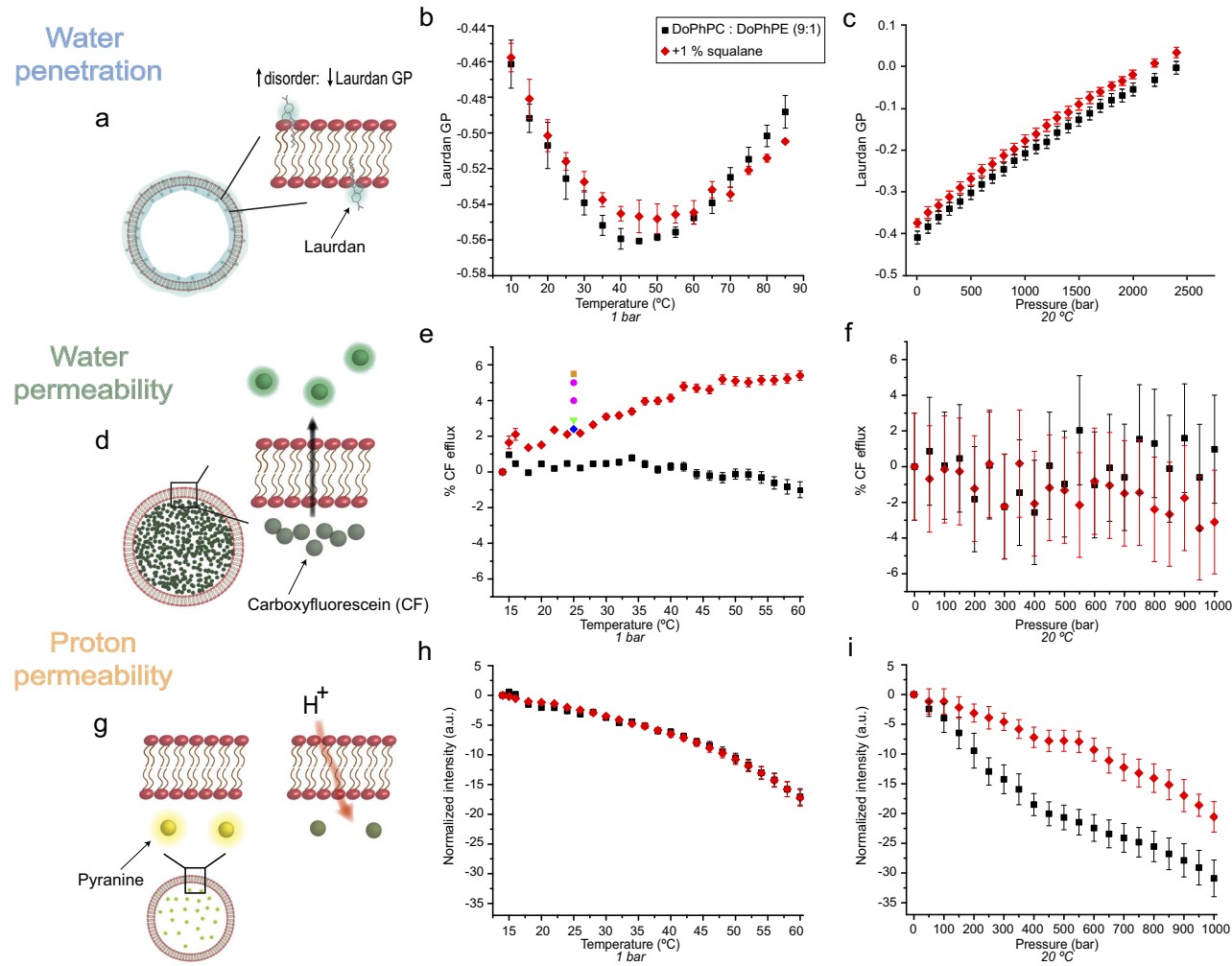

**Fig. 5 Squalane modulates membrane physico-chemical properties.** Water penetration in the headgroup region of the bilayer measured by laurdan GP (**a–c**), water permeability of lipid bilayers measured by carboxyfluorescein (CF) efflux (**d–f**), and proton permeability of lipid bilayers measured by pyranine fluorescence (**g–i**) as a function of temperature (middle column) and pressure (right column) for DoPhPC:DoPhPE (9:1) (black spheres) and in the presence of 1 mol% squalane (red triangles). Dots in the temperature-CF efflux data represent an approximation of the %CF efflux for different lipids at 25 °C: black square: 1,2-dioleoyl-*sn*-glycero-3-phosphocholine, red spheres: 1-palmitoyl-2-oleoyl-glycero-3-phosphocholine, green triangle: 1,2-dimyristoyl-*sn*-glycero-3-phosphocholine, and blue diamond: 1,2-diphytanoyl-*sn*-glycero-3-phosphocholine[62–65].

to favor the formation of bilayers and obtain unilamellar liposomes, which are necessary for permeability measurements. These unilamellar vesicles are not in close contact with one another, and thus the formation of non-lamellar phase is much more unlikely. Furthermore, the very large excess of water induces a strong constraint on the lipid system, so it is difficult to escape the bilayer phase. Indeed, vesicles disappearance from the sample normally yields lipid micelles rather than non-lamellar phases.

Laurdan is a fluorescent probe with a long hydrophobic tail sensitive to its water environment. Hence, it will report on lipid packing and phases[59] depending on how deep it is inserted in the membrane. When localized in a less polar environment, the probe emission is blue. It becomes green when placed in a more polar environment. Thus, the measured value of its GP reports the polarity of its surrounding environment. Positive laurdan GP values indicate an ordered phase into which laurdan penetration is more limited. GP values of about 0.6–0.8 are typical of a gel phase. Negative laurdan GP values are indicative of a fluid phase. The typical liquid-crystalline phase which is the functional state of the plasmic membrane has GP values of −0.4 to −0.2[56]. Here, the blue-shifted emission of laurdan observed at high

temperatures could be explained by the relocation of the probe due to the slow phase transition. By causing a higher disorder in the lipid bilayer, increasing temperature may lead to a reorganization of the polyisoprenoid chains of the lipids, leading to the insertion of the naphthalene rings of laurdan deeper into the lipid bilayer. The higher laurdan GP values in the presence of 1 mol% squalane indicate a slightly higher rigidity of the membrane (Fig. 5c). Additional Fourier-transform infrared spectroscopy (FT-IR) experiments were performed to yield information on the population of lipid conformers of the membrane. The FT-IR data showed only minor changes upon addition of the apolar intercalant (Supplementary Figs. 2–3) also supporting a minute increase in the rigidity of the membrane. The congruence of the two types of analysis gives further support to the role of squalane in increasing the rigidity of the membrane, although to a much lesser extent than expected by the theoretical model. How this increased rigidity plays a role in the adaptation to the variable hydrothermal vent environment is yet to be explored, but we believe that by allowing the membrane to adopt negative curvature, squalane may play a decisive role in the maintenance of the overall membrane structure limiting the deleterious effects of fluctuations of environmental variables.

Large-molecule leakage was probed by the carboxyfluorescein (CF) release assay, which is a well-established method that monitors the increase in fluorescence caused by the dilution of the self-quenched CF upon release from leaky liposomes into the surrounding medium[53,60] and thus probes the permeability of lipid bilayers to solutes. By extension, it is often used as a proxy of the leakage of molecules that do not use the proton channel to cross the membrane. This is used to mimic the transfer of water through the lipid layers. Thus, it does test the intrinsic impermeability of the lipid layers, which needs to be high enough in order for the cell to control its inward and outward water flows, which occurs in cells by the mediation of transmembrane proteic water channels. Temperature and hydrostatic pressure (Fig. 5d–f) have contrasting impacts on squalane-containing DoPhPC:DoPhPE membranes. The presence of squalane increases the leakage of CF from liposomes upon an increase in temperature, demonstrating an increase in membrane permeability (Fig. 5e). Such a tendency has also been reported when other isoprenoid molecules, such as dolichol, were present in the lipid bilayer[61]. However, the presence of squalane did not impact CF release as a function of the pressure range covered (Fig. 5f). These results demonstrate that the presence of squalane in the lipid bilayer influences the essential physico-chemistry of the membrane bilayer. Most importantly, the T-induced increase in water permeability allows permeability to reach a level that is essentially similar to that of bacterial-like lipid systems (color dots in Fig. 5e) also tested in the absence of water channels[62–65], while the intrinsic permeability of the membrane in the absence of squalane is extremely low, as expected for archaeal lipids, and may be too low from a biological point of view to allow a functional membrane[66].

The membrane influx of protons into liposomes was measured by monitoring the emission of pyranine, a pH-sensitive fluorophore (excitation 470 nm; emission 510 nm)[67] (Fig. 5g) that is widely used to study the internal pH of liposomes. Interestingly, no influence of the apolar intercalant on membrane proton leakage was observed upon increasing the temperature (Fig. 5h). In contrast, proton permeability into vesicles upon an increase in pressure is strongly decreased when squalane is present (Fig. 5i). Such an impact of the intercalant was predicted by different molecular models for the proton leakage of lipid bilayers[22]. With squalane present, increasing pressure increases the packing density of the lipid bilayer, as can also be deduced from lattice parameter values (Supplementary Figure 1), and the presence of squalane in the center of the bilayer will further block proton transport. As a consequence, the proton motive force will be enhanced in the membranes containing squalane, as will be the tolerance to environments with a huge abundance of protons, i.e. acidic environments.

The usual decrease of proton permeability of bacterial lipids-based synthetic membranes at high hydrostatic pressure is mostly due to an increase of chain ordering and therefore an increase of the lipid bilayer thickness[68], which is inversely proportional to proton permeability[69]. In contrast, archaeal polyisoprenoid chains are already tightly packed and ordered under ambient pressure conditions. The lipid bilayer formed by DoPhPC:DoPhPE (9:1) does not show a significative increase of bilayer thickness with increasing hydrostatic pressures. Thus, it is not unexpected that an increase in hydrostatic pressure on this membrane system may not lead to a decrease in proton permeability as seen for bacterial/eukaryal lipids. Moreover, the data acquired by different methods in this study does not show evidence of an increased packing of the isoprenoid chain. In contrast, we observed an increase in the bilayer thickness upon insertion of squalane, which would be congruent with the observed decrease of proton permeability observed here.

It may appear counter-intuitive that membrane permeability to proton and solutes do not follow the same P and T trends. Membrane permeability to water is highly correlated to membrane fluidity and it depends on lipid solubility as well as on the area per lipid ratio. Several models have been proposed to explain such permeability: "the one-slab model"[70], "the three-slab model"[71] and "the channel model"[72]. However, all models proposed to explain solute permeability fail when referring to proton permeation, which correlates weakly with fluidity, but exponentially and with inverse proportionality to membrane thickness[69]. For instance, the liquid ordered phase achieved in the presence of cholesterol decreases water permeability but increases proton permeation[73]. In the case of large uncharged molecules/water, we believe that by releasing chain frustration, squalane induces the lateral membrane reorganization into domains corresponding to the two coexisting phases that are visible in laurdan. This coexistence would generate domain boundaries of imperfect packing, e.g. cracks, through which large molecules could circulate. The higher the number of domains, the higher the number of imperfections, the higher the permeability. In contrast, pressure does not seem to modify the equilibrium between the two phases, potentially explaining why permeability is not affected. The contrasting effect observed for proton permeability clearly shows that large molecules and protons do not cross the membrane by the same mechanisms and possibly not at the same location. For both types of permeation, we believe that the complete rationale for these observations will only be found when the path of each type of molecules through the archaeal membrane will be clearly characterized.

**Apolar lipids may help maintaining membrane functionality under combined high temperature and pressure.** In the membrane models proposed by Cario[21] or Haines[22], the presence of apolar intercalants within the membrane is expected to affect the stability, but more importantly, to shift the functionality domain of the membrane towards higher temperatures and hydrostatic pressures. Thus, apolar 'lipids' would represent one of the routes by which organisms could adapt to high-temperature and high-pressure environments without requiring the synthesis of bipolar, membrane-spanning, and monolayer-forming lipids.

The apolar lipid squalane in the archaeal membrane analog have been localized within the T and P ranges 25–85 °C and 1–1000 bar, respectively, covering a large majority of the temperature and pressure conditions encountered in nature by *T. barophilus*. For each condition, we could demonstrate that squalane remained inserted in the center of the midplane, parallel to the surface of the membrane, similar to the control conditions at room temperature and ambient pressure (Supplementary Fig. 4). From the intensity of the NSLD data, we can infer that the amount of squalane inserted within the membrane is similar for all temperatures and pressures. The d-spacing of the membrane in the absence of the apolar molecule was essentially insensitive to pressure and temperature (Supplementary Figs. 5 and 6) with a variation in d-spacing between the two most extreme conditions of only 0.2 Å. In contrast, the d-spacing of the membrane containing 1 mol% squalane decreases with increasing pressure from 54.6 to 53.1 Å (Supplementary Fig. 5), indicating that squalane renders the archaeal membrane analog pressure sensitive and significantly more laterally compressible. As expected, the compressibility increases with increasing temperature (Supplementary Fig. 6). Notably, the d-spacing of the membrane containing squalane at 85 °C and 1000 bar is very similar to that of the membrane without squalane at room T and ambient P (Fig. 6). This finding resembles the "corresponding state principle" that was first formulated by Vihinen[74] and

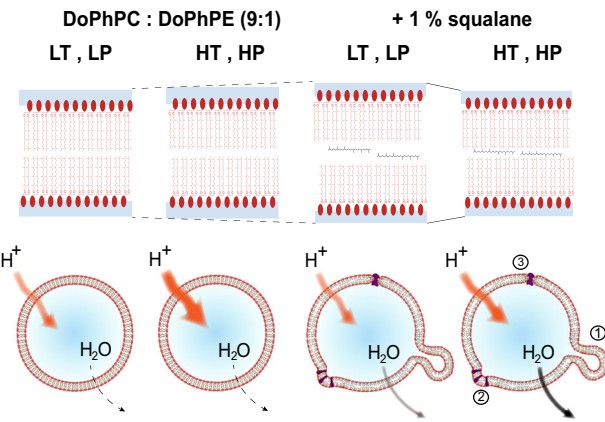

**Fig. 6 Schematic view of the impact of apolar intercalant on membrane structure and function.** The presence of apolar molecules renders the membrane more responsive to pressure, modulates its permeability and facilitates membrane curvature. LT: 25 °C, LP: 1 bar, HT: 85 °C, HP: 1000 bar. Top: effects of temperature, pressure, and squalane on the lipid bilayer. In the absence of squalane, a slight increase in lamellar repeat spacing is observed, which is due to the volume expansion. At LT and LP in the presence of squalane, the lamellar repeat spacing increases due to an increase in the hydrophobic region of the lipid bilayer. The membrane containing 1 mol% squalane is rather compressible, leading to a decrease in the d-spacing. Bottom: representation of the permeability of the lipid bilayer in the absence and presence of 1 mol% squalane at two extreme conditions: 25 °C (LT), 1 bar (LP) and 85 °C (HT), 1000 bar (HP). In the absence of squalane, the lipid bilayer membrane is slightly impermeable to water efflux. However, its proton permeability is high and increases considerably at high temperatures. When 1 mol% squalane is added, the lipid vesicles display a higher permeability towards water efflux, similar to that of typical bacterial lipids. However, its proton influx is lower. Furthermore, the presence of squalane induces bending membrane modifications that allow the cell membrane to adopt essential curvatures for different cell functions, such as the formation of vesicles (1), insertion of membrane proteins that need specific curvature (2), and transmembrane proteins anchored in lipid bilayers (3).

Jaenicke[75], stating that the flexibility of a protein adapted to extreme conditions should be the same as the flexibility of a protein adapted to mesophilic conditions under ambient conditions. Such corresponding state principle was previously reported for proton permeation in archaeal monolayer membranes[11,66]. Here, conclusions could be drawn the structural characteristics of membrane analogs functioning under normal conditions but adapted to extreme conditions. Increasing compression of lipid membranes is usually associated with a transition to an ordered phase, an interdigitation of the lipids' hydrophobic chains, and/or to a tilt of the molecules at an angle perpendicular to the surface, which allows for a more compact lipid bilayer[76]. In the present case, compression of the membrane does not induce a phase transition to a gel phase or, according to the diffraction data, to an interdigitation of the isoprenoid chains, as evidenced by the calculated dimensions of the hydrophobic domain of the membrane ($2D_c$, Fig. 2). Therefore, under all P and T conditions tested, the archaeal membrane analog remains in a fluid-like, liquid-crystalline, and hence functional state, which is congruent with previous observations on archaeal membrane permeability to protons.

The membrane analog structure at the highest temperature and pressure tested suggests a possible role of squalane in modulating membrane fluctuations at the temperatures that hyperthermophilic Archaea, such as *T. barophilus*, experience in hydrothermal vent environments. Our results on a simplified archaeal

membrane analog show a loss of membrane organization at temperatures above 70 °C in the absence of squalane (Fig. 3 right). In contrast, in the presence of 1 mol% squalane, which correspond to the proportion of apolar polyisoprenoid in the membrane of *T. barophilus*, both lamellar and non-lamellar phases appear stable at all temperatures studied. The compressibility data of the membrane analog suggests that a second role of squalane may be to keep the membrane impermeable to proton flux, i.e., to help sustain the proton motive force of the membrane and reduce water influx across the membrane under high hydrostatic pressures (Fig. 6). Third, our SAXS data strongly associate the presence of squalane with the ability of the archaeal membrane analog to adapt its membrane curvature (Fig. 3). The clustering of non-lamellar structure-forming lipids is essential in several transient cellular processes, such as membrane fusion and fission (Fig. 4), since it would help to decrease the bending membrane energy. For the archaeal lipid analogs studied here, we did not observe these large negative curvatures in the absence of squalane, even at the most extreme P and T conditions tested. This suggests that without apolar intercalants, the classical amphiphilic archaeal lipid mixtures modeled by our system might also lack the ability to modulate membrane bending under the physiological conditions of the archaeon. Furthermore, members of the order Thermococcales, including *T. barophilus*, have been shown to produce large quantities of vesicles, often hundreds per cell[77]. Although it is not completely clear what role these vesicles play in the archaeal life cycle, they are proposed to be involved in genetic exchange or in the interference of the growth cycle of competing species[78,79]. Regardless of their function, the production of hundreds of vesicles per cell highlights the ability of the natural membrane to adapt its curvature during the intermediates of the membrane fusion/ fission processes[40], for which apolar intercalants embedded in the membrane may be facilitators.

Our results confirm and expand on the hypotheses discussing novel membrane architectures as proposed by Cario and colleagues[21]. Importantly, we provide an experimental demonstration that in this membrane architecture, the lipid bilayer stability and functionality are shifted to higher temperatures under high pressure, which supports the view that apolar lipids may constitute one of the adaptive routes to high-temperature tolerance in that could exist in archaeal hyperthermophiles. In the future, it will be important to further explore the relative contribution of the different polar and apolar lipids on membrane parameters as a function of hydrostatic pressure and temperature, since increasing hydrostatic pressure and temperature have antagonistic effect on molecular systems, including lipids. Indeed, for instance increasing temperature will increase molecular motion, while increasing hydrostatic pressure will decrease it. Thus, the presence of the apolar lipid might be play an important role in the fine regulation of membrane properties under combined high pressure and temperature.

In this study, we could only model a very simplified analog of an archaeal lipid bilayer and thus, common features of natural membranes (e.g. variety of lipids, presence of membrane proteins) and specific ones of archaeal membranes (glycerol stereochemistry) have not been tested. In particular, the role of membrane-bending proteins on membrane remodeling cannot be ignored in Archaea, as shown in Asgard archaea in which the particular morphology of *Prometheoarchaeum syntrophicum* with extensive membrane protrusions is most likely caused by membrane-remodeling ESCRT-like proteins. Hence, if demonstrated, the ability of apolar polyisoprenoid will not be the only means by which Archaea might introduce high bending in their membranes, but might play a role in specific instances in which membrane proteins are not favorable. In addition, the presence of

apolar polyisoprenoids in archaeal membranes still needs to be explored in depth, since it has not been searched for nor quantified in most of them[51]. Taking this into account, here we made a first experimental validation of a molecular model that may help to understand alternate adaptive routes to membrane adaptation to high temperature and support the structural and functional stability of the membrane of the extremophilic Archaea lacking the ability to synthesize monolayer-forming membrane lipids. Most importantly, the membrane model architecture explored here has possible implications in the way we understand the evolution of membrane adaptation to extreme conditions, especially with regard to the role of the bipolar lipids[20,80]. It may also help reconsider the apparent contradiction raised by the presence of bipolar lipids in mesophilic environments, such as the open ocean, or mesophilic Archaea such as Thaumarchaeota, which did not fit with the proposed temperature adaptation dogma. This study opens new perspectives to expand the model for the cell membranes of bacteria, where apolar intercalants may also be present and shown to intercalate in the same location as in archaeal lipid-based membranes[23]. Last, this architecture could be transposed to the membrane structure of the first cells[81–83], which would provide further support to an origin of life in the variable and extreme environment of hydrothermal vents. It will be crucial to test whether apolar molecules can be inserted within the midplane of protolipid bilayers, such as those produced from decanol and decanoic acid[84], and see whether the stability and functionality domains can be extended to higher or more variable temperature conditions.

A final important prediction of the novel membrane ultrastructure of Cario and colleagues[21] concerns the spatial disposition of the mono- and bipolar lipids in the membrane. Indeed, in contrast to *M. kandleri* which produces only monopolar lipids, *T. barophilus* produces between 15% and 50% bipolar, monolayer-forming, lipids, depending of growth conditions[20,21]. Under these conditions, there is a high probability of the coexistence of membrane domains of diverging lipid compositions. Our results suggest that apolar molecules such as squalane favor phase separation in the membrane (Supplementary Fig. 7) and thus may promote domain formation in native cells. Such distinctive domains must have different physico-chemical properties and therefore different functions related to, for example, fusion, enzymatic activity or membrane protein docking[85]. Phase separation has been shown to exist in reconstructed archaeal membranes made solely of monolayer-forming lipids[86]. Further work will be required to identify the existence of these mono/bilayer domains in natural membranes in *T. barophilus* and identify the proteins specific to each domain type to decipher their role in the cell cycle and the adaptation to the environment.

## Methods

**Chemicals**. The synthetic archaeal-like lipids 1,2-di-O-phytanyl-*sn*-glycero-3-phosphocholine (DoPhPC) and 1,2-di-O-phytanyl-*sn*-glycero-3-phosphoethanolamine (DoPhPE), were purchased from Avanti Polar Lipids Inc. (Albaster, USA). The apolar isoprenoid used, 2,6,10,15,19,23-hexamethyltetracosane (squalane), was purchased from Sigma-Aldrich (Saint-Louis, USA) in its hydrogenated form and from CDN Isotopes (Pointe-Claire, Canada) in its deuterated form. All other chemicals, including 8-hydroxypyrene-1,3,6-trisulfonic acid trisodium salt (pyranine), 5(6)-carboxyfluorescein (CF), and 6-dodecanoyl-N,N-dimethyl-2-naphthylamine (laurdan) were purchased from Sigma-Aldrich (Saint-Louis, USA).

**Neutron diffraction experiments**. Contrast variation neutron diffraction is underpinned by the difference in coherent neutron cross sections of $^1$H and $^2$H (deuterium, $D$)[87], which is widely used to highlight parts of a sample. Indeed, $D$ is a much stronger coherent scatterer (scattering length 6.7 fm) compared to $H$ ($-3.7$ fm)[88]. Thus, profiles of the scattering length densities (i.e., the membrane's cross-sectional profile), calculated as the Fourier sum of the structure factors determined

by neutron scattering, show increased density at the location of the deuterated molecule.

Neutron diffraction experiments were performed on multiple stacks of oriented membrane bilayers. Bilayers were prepared by spreading 3 mg of DoPhPC:DoPhPE (9:1 molar) in chloroform:methanol (2:1) with or without 1 mol% squalane on a silicon wafer and dried under vacuum. Diffraction patterns were measured on the D16 small-momentum transfer diffractometer (D16[31]) at the Institut Laue Langevin (Grenoble, France), and we used a cryostat and high-pressure equipment[89,90] to precisely control temperature and pressure, respectively. We used an incident wavelength of $\lambda = 4.5$ Å and $2\Theta$ scans with an accessible q-range from 0.06 Å$^{-1}$ to 0.51 Å$^{-1}$.

To locate squalane within the membrane system, we made use of the scattering density difference between hydrogen and its isotope deuterium using protonated and deuterated squalane as intercalants in the synthetic membranes[23]. This difference makes it possible to determine the position of a deuterated target molecule. The diffraction patterns were measured up to the fourth order of the Bragg reflection of the neutrons scattered by the multistack of lipid bilayers. The integrated intensities of the Bragg peaks were corrected for absorption and analyzed using Gaussian functions (with a maximum error of 5%), which provided a suitable model for describing the shape of the Bragg reflection, and up to four orders of Bragg reflections were taken into account for subsequent analysis.

The sum of neutron scattering lengths per unit volume is known as the neutron scattering length density (NSLD) profile. The NSLD profiles were calculated as discrete sets of Fourier coefficients $f_n$ according to the formula:

$$\rho_{bilayer}(z) = \frac{2}{D}\sum_{n=1}^{M} f_n \nu_n \cos\left(\frac{2n\pi}{D}z\right)$$

where $D$ is the lamellar spacing of the bilayers in the $z$ direction (perpendicular to the lipid interface; $z \in \left[\frac{-D}{2};\frac{D}{2}\right]$. Coefficients $f_n$ can be obtained by $I_n = \frac{|f_n|^2}{Q_z}$, $Q_z$ is the Lorentz correction factor equal to $q$ for oriented bilayers, $I_n$ is the integrated intensity of the $n^{th}$ Bragg peak and $\nu_n$ corresponds to the phase of the structure factor.

The lamellar d-spacing of the lipid bilayer was determined from the observed $2\Theta$ of the Bragg peak positions according to Bragg's law, $n\lambda = 2d\sin\Theta$, where $n$ is the diffraction order and $\lambda$ is the selected neutron wavelength. To obtain the phases of the structural factors, each type of squalane sample was measured at four different D$_2$O-H$_2$O contrasts (D$_2$O content: 8%, 20%, 50%, 100%). Therefore, it was possible to use the linear correlation of the structure factor amplitudes and sample D$_2$O content[91] to yield the structure factors for each sample.

The membrane structural parameters were defined at 25 °C and 100% relative humidity as follows: A: average interfacial area/lipid; $V_L$: lipid molecular volume, $V_L = V_H + V_C$; $V_H$: volume of the head group; $V_C$: sum of volumes of chain methyls; $D$: lamellar repeat spacing; $D_{HH}$: headgroup peak-peak distance, $D_{HH} = 2(D_C + D_{H1})$; $D_C$: thickness of hydrocarbon core, $D_C = V_C/A$; $D_B$: Gibbs-Luzzati bilayer thickness, $D_B = 2V_L/A$; $D_w$: Gibbs-Luzzati water thickness, $D_w = D - D_B$; $D_{B'}$: steric bilayer thickness, $D_{B'} = 2(D_C + D_{H'})$; $D_{w'}$: steric water thickness, $D_{w'} = D - D_{B'}$; $D_{H'}$: steric headgroup thickness, $D_{H'} = (D_B/2) - D_C$; $D_{H1}$: partial headgroup thickness, $D_{H1} = (D_{HH}/2) - D_C$; $n_w$: number of water molecules/lipid, $n_w = AD_w/2V_w$ ($V_w$: water molecular volume, ~30 Å$^3$); and $n_{w'}$: number of water molecules between $D_C$ and $D_{B'}/2$, $n_w = (AD_{H'} - V_H)/V_w$.

**Small-angle X-ray diffraction**. Small-angle X-ray diffraction (SAXS) was carried out on fully hydrated lipid solutions prepared from 5 mg of the DoPhPC:DoPhPE (9:1 molar) mixture in 20 μL of water with or without 1 mol% of squalane placed inside a pressure chamber[92]. The pressure- and temperature-dependent experiments were carried out at six different temperatures (10–85 °C, in 15 °C steps) and from 1 to 1000 bar (in 50 bar steps). Experiments were performed at beamline I22 of the Diamond Light Source (Didcot, United Kingdom) with an energy of 18 keV. The momentum transfer was defined as $q = 4\pi \sin(\Theta)/\lambda$, where $2\Theta$ is the scattering angle. The lattice parameters are given by $a = 2n\pi/q$ and $a = 2(2n\pi/q)/\sqrt{3}$ for a lamellar and a hexagonal phase, respectively, where $n =$ order of reflection. The type of phase can be distinguished by the characteristics of the SAXS peak ratios for the lamellar phase 1:2:3:4, the cubic Pn3m phase $\sqrt{2}:\sqrt{3}:\sqrt{6}$ and the hexagonal phase 1:$\sqrt{3}$:2:$\sqrt{7}$.

**LUV preparation**. Lipid unilamellar vesicles (LUVs) from DoPhPC: DoPhPE (9:1 molar) were prepared by dissolving the lipids with the appropriate amount of squalane in chloroform/methanol (2:1 v/v) to obtain 1 mol% squalane, vortexing the solution and drying it under a stream of nitrogen gas. The dried solution was left to dry overnight under vacuum. The lipid film was rehydrated with buffers specific to each technique, followed by vortexing and five cycles of freezing/thawing. LUVs were formed by pressure extrusion[93] passing the solution 11 times at 60 °C through a 100-nm polycarbonate membrane using a Mini Extruder® (Avanti Polar Lipids, Inc., USA). After extrusion, the solution was cooled down, and the excess dye was removed by chromatography of the LUVs over Sephadex G-50 M columns. LUVs were used immediately after the chromatography.

**Laurdan generalized polarization**. Laurdan is a fluorescent probe whose emission wavelength is determined by water penetration into the lipid headgroup region, *i.e.*, is sensitive to the phase state of the membrane. Laurdan was excited at 350 nm, and its emission was measured between 420 and 510 nm. Temperature-dependent measurements were performed using a quartz cuvette with a volume of 100 μL in the temperature range 10 to 85 ± 0.1 °C. The spectral changes of laurdan are quantified by the generalized polarization function, GP[94], defined as

$$GP = \frac{I_{440} - I_{490}}{I_{440} + I_{490}}$$

where $I_{440}$ and $I_{490}$ are the emission intensities at 490 and 440 nm, which are characteristic of fluid (liquid-crystalline) and ordered (gel) lipid phase states, respectively. Hence, the GP value reports on the fluidity of the membrane. Laurdan GP measurements were performed on LUVs, which were prepared from a lipid mixture containing 0.2 mol% of the fluorescent probe laurdan. The final lipid concentration in the LUV suspension was adjusted to 1 mM. Temperature-dependent fluorescence spectroscopic measurements were performed on a K2 multi-frequency phase and modulation fluorimeter (ISS, Inc., USA).

**5-(6)-Carboxyfluorescein efflux**. LUVs were prepared in a buffer (HEPES 10 mM, 100 mM KCl, 1 mM EDTA pH: 7.8) containing 40 mM CF. At this concentration, CF is self-quenched[60], and the release from the liposomes will increase its fluorescence intensity. The final lipid concentration in the LUV suspension was adjusted to 5 mg/mL, and the fluorescence was monitored at 518 nm (excitation at 492 nm) as a function of temperature (10–60 °C) and pressure (1–1000 bar) at 20 °C. At the end of each measurement, a 100% reference point was determined by adding 0.1% Triton X100 to disrupt the liposomes, facilitating complete release of the remaining trapped fluorescent dye. The results are presented as the ratio[95]:

$$\%CF_{efflux} = \frac{F_t - F_0}{F_{max} - F_0} \times 100$$

where $F_t$, $F_0$, and $F_{max}$ are the fluorescence intensities at time t, time zero, and after total solubilization by Triton X100.

**Pyranine fluorescence**. The membrane influx of protons into liposomes was measured by monitoring the emission of pyranine, a sensitive pH sensor[67]. LUVs were prepared from a 5 mM HEPES buffer (pH = 7.5) containing 5 mM pyranine, and the final lipid concentration was adjusted to 5 mg/mL. Measurements were performed as a function of temperature (10–60 °C) and pressure (1–1000 bar) at 20 °C in the presence of 0.1 M HCl as a proton source. All results are normalized to a blank experiment performed under the same conditions in the absence of an external proton source to take care of a possible impact of P and T on the photophysics of the reporter dye. The excitation wavelength was 470 nm and emission was read between 500 nm and 520 nm with its maximum at 510 nm.

**Statistics and reproducibility**. The neutron diffraction results represent three biological replicates, each made of three technical replicates (three different D₂O/H₂O contrasts). The SAXS data represent two biological replicates, each made of three technical replicates. The water penetration, water permeabilty, and proton permeability data represent triplicate (Temperature) or duplicate (high hydrostatic pressure) measurements. No data points were excluded from the analyses.

**Reporting summary**. Further information on research design is available in the Nature Research Reporting Summary linked to this article.

## Data availability

The data that support the findings of this study are available from the corresponding authors upon request. Data obtained at ILL are identified by https://doi.org/10.5291/ILL-DATA.8-02-809 and https://doi.org/10.5291/ILL-DATA.8-02-818.

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

## Acknowledgements

We gratefully acknowledge the Institut Laue-Langevin and the Diamond Light Source for the allocation of beam-time, and we thank O. Aguettaz, C. Payre, and J. Maurice for their technical support on these instruments. We thank the research team Chimie Organique et Bioorganique (UMR 5246, INSA Lyon) and particularly Pr. S. Chambert for access to their spectrofluorometer. This work was supported the CNRS program for inter-disciplinary studies Origines, by the French National Research Agency program ANR 2017 BLAN 17-CE11-0012-01 (ArchaeoMembranes) to P.M.O. and J.P., the German-French bilateral research cooperation program "Procope" 2018–2019 to J.P. and R.W., and the Royal Society to P.M.O. and N.J.B. M.S.C. was supported by a PhD grant from the French Ministry of Higher Education and Innovation.

## Author contributions

P.M.O., J.P., R.W., N.J.B., and M.S.C. designed the experiments. M.S.C., P.M.O., and N.E. performed the experiments and acquired the data. M.S.C., M.G., B.D., J.P., R.W., and N.J.B. analyzed the data. All authors took part in the writing of the manuscript. All authors agree with the publication of the manuscript in its current form.

## Competing interests

The authors declare no competing interests.
