## [Peer Review File · Communications Biology]

Reviewers' comments:

Reviewer #1 (Remarks to the Author):

In the manuscript „A novel membrane ultrastructure explains the cell adaptation to extreme conditions in Archaea” by Castell et al., the authors report the effect of the insertion of squalene into an archaeal lipid bilayer analogue. First they address the insertion of the squalene in the bilayer. Later on, they study the effect of the squalene insertion on the structure and function of the artificial membrane by analysing for example membrane curvature as well as water, solute and proton permeability in response to temperature and pressure change.

The studies were performed by using small-angle X-ray scattering (SAXS), neutron diffraction experiments as well as spectroscopic experiments with unilamellar lipid vesicles using fluorescent dyes.

With their artificial system they show the insertion of the squalene in the midplane of the bilayer increase the permeability of water and reduce the permeability of protons under extreme conditions.

In addition, apolar isoprenoids induce a negative curvature in the membrane.

The authors represent a thorough and interesting analysis that fits nicely in the current discussion of the origin of life. However, in order to be of relevance for a broader public the author's should consider to give more details about the current understanding of the archaeal membrane composition and clearly state the differences to their artificial system (for further details see below).

L20 ff (Abstract) In the abstract the performed experiments and major findings of this study are not clearly presented. Maybe it would be helpful to include their experiential approach here.

L44 ff (Introduction). Here the authors have to present a more thorough presentation about the current knowledge of the archaeal membrane in order to allow the comparison to their artificial model. Thus, it has to be stated that the polar head group is formed by glycerol 1-phosphate (not glycerol-3-phosphate). What are the phosphate-bound substituents in Archaea and of interest for this study of *Methanopyrus kandleri*?

In this regard it might be also interesting that membrane fluidity in Archaea is regulated by the introduction of heterocycles whereas unsaturated fatty acids are found in Bacteria and Eukaryotes. Finally, as in Bacteria and Eukaryotes numerous proteins are found in the membrane.

L63 The authors might want to give the growth range and growth optimum.

L84ff This passage is rather speculative. It seems to be more appropriate for a final discussion and in my opinion the authors need to be very clear and careful what conclusions they can retrieve from their artificial archaeal membrane model.

L91/92 The sentence should be rephrased to: Last, the architecture also implies the existence of membrane domains of divergent compositions “in the archaeal membrane”,....

L113 To what extent does the model represent the bilayer structure of *Thermococcus barophilus*? The authors might want to include why they decide on the chosen archaeal-like lipids DoPhPC and DoPhPE. Was it due to the commercial availability?

L128 The lycopane structure might be shown in comparison to the squalene structure in Fig. 1.

L128 The Fig 2B needs to be explained in more detail. What is shown here?

L137 From the results shown in Fig 2C can it be excluded that the squalene intercalates partially into both monolayers and thus connects them to a monolayer. I am not familiar with the respective methods, but this is what I would expect from hydrophobic molecules and the NSLD observed might fit approximately to the size of the squalene. Also, the author's should discuss if the preparation method might result in artefacts.

L171ff For me the phase discussion it is a bit hard to follow. May be the authors can explain initially what kind of different phases and transitions can be observed using their experimental approach.

L256ff The authors might want to mention that they use unilamellar liposomes for the spectroscopic permeability assays.

L282ff As far as I know water is typically transported via membrane proteins (water channels, aquaporines) in living cells. These have been also reported for Archaea e.g. <https://www.pnas.org/content/102/52/18932>.

L376ff In the discussion the possible roles of proteins that can be not addressed using the archaeal membrane model needs to be considered. Besides aquaporines for water transport also proteins are discussed to be involved in vesicle formation for example the ESCRT system in Crenarchaeota, FtsZ-based system in Euryarchaea <https://www.nature.com/articles/nrmicro2406>.

L392ff There is quite some insight into the role of vesicles in Archaea.

L403ff In the final discussion it should be clarified that this is only a model and that typical features for the archaeal membrane (see discussion above) are missing.

L465 Fig. 2 D The model shows glycerol 1-phosphate (see fig. 1) in the experiment glycerol 3-phosphate is used a head group.

Reviewer #2 (Remarks to the Author):

This paper describes an experimental study of model membranes composed of diether phospholipids with or without the apolar isoprenoid squalene at the midplane. Interest in this membrane architecture is motivated by the relative abundances of diether versus tetraether lipids in some extremophile Archea membranes. The rationale for employing synthetic lipids in this study is clearly presented and convincing. The authors employ neutron scattering, small-angle X-ray scattering, and permeability assays to characterize these model membranes at different temperatures and pressures and to assess how the presence of the apolar intercalant might contribute to the adaptation of organisms to extreme conditions.

The reported observations are of high quality and support the interpretations offered by the authors. I think that this is a significant and novel contribution. It provides some very interesting insights into a membrane architecture that appears to enable some organisms to maintain functional membranes over wide ranges of temperature and pressure. It also poses some interesting questions about aspects of this class of membrane, particularly its response to applied pressure.

The comparison of neutron scattering length densities for the diether phospholipid bilayers containing either hydrogenated or deuterated squalene provide an unambiguous demonstration of squalene's location at the bilayer midplane. Small-angle X-ray scattering was used to characterize phases observed over a range of temperatures at ambient pressure and at 1000 bar for bilayers without and with squalene. The insensitivity of d-spacing to pressure in the absence of squalene as shown in Figure 3 and in Figure S1 is striking. I would have been interested in a bit more detail about the two lamellar phases found to coexist in the absence of squalene. The presence of squalene gave also promoted the coexistence of a cubic phase with either lamellar or hexagonal phases and increased the sensitivity of the lamellar and hexagonal structures to pressure. The authors reasonably attributed the

accommodation of non-lamellar organization to structural flexibility provided by the presence of squalene at the midplane.

The authors used laurdan generalized polarization (GP) to assess chain order (as a proxy for water penetration into the model membranes), carboxyfluorescein (CF) efflux to assess permeability to larger molecules (and as a proxy for permeability to water), and pyranine fluorescence to assess permeability of the membranes to protons, presumably as H_3O^+ . These assays were done as functions of temperature and pressure in LUV bilayers without and with squalene. I was unable to find a statement about the temperature at which the pressure dependent LUV studies were carried out but based on comparison of the ambient pressure results from panels C, F, and I of Figure 5 with the corresponding temperature dependent observations in panels B, E, and H, I am guessing that the pressure experiments were done at about 10°C . It would be helpful to have some comment about the rationale for this choice of temperature for the LUV experiments and whether the pressure dependent behavior might have been different at higher temperature since high temperature and high pressure might be an interesting condition to probe for this system. If there was an experimental constraint on temperature for the high pressure LUV experiments, a comment would be appropriate.

The laurdan GP observations suggest that squalene has a small ordering effect on the bilayer. Squalene does not substantially alter the sensitivity of the laurdan GP to changes in temperature or pressure. This is probably not too surprising since laurdan is not sampling the membrane environment close to the midplane. The interpretation of the laurdan GP observations seems to be based on existing results for laurdan in bilayers of diester lipids. The authors might want to consider how differences between hydration of the diether and diester lipids might be reflected by laurdan GP. There is some discussion of the effects of ether versus ester linkage on laurdan GP in a paper by Pérez et al. [Pérez et al. (2021) Effect of cholesterol on the hydration properties of ester and ether lipid membrane interphases. *Biochim. Biophys. Acta* 1863, 183489]. The Pérez et al. paper appeared online only recently, presumably after the current manuscript was submitted, but maybe there is some earlier literature on laurdan GP in diether lipids that might be pertinent. I don't think that differences in laurdan GP due to the ether linkages would alter the main findings that squalene has only a small effect on order and that order increases with pressure but some additional context would be helpful.

Panel E of Figure 5 shows that squalene increases the permeability of the vesicle membranes to CF and thus, presumably, to molecules like water. The authors reasonably attribute the squalene-induced increase in CF efflux with increasing temperature to phase coexistence and domain boundaries. The observation, in panel F of Figure 5, that squalene does not change the sensitivity of CF influx to pressure is not too surprising. If this experiment was done at a temperature corresponding to the lowest temperature in panel E of Figure 5, that would be a temperature at which CF efflux is already low and largely independent of the presence of squalene. Increasing the pressure at this temperature would likely keep the efflux at a low level (possibly reflected by the large error bars in panel F).

The effects of temperature and pressure on the permeability to protons in the absence and presence of squalene, shown in panels G, H, and I of Figure 5 is quite interesting. While panel G does illustrate the connection between increasing permeability to protons and decreasing intensity, it might not hurt to explicitly mention that relationship in the text or in the figure caption. The authors note that squalene has no effect on proton permeability at ambient pressure which is indeed surprising. It is particularly interesting, though, that increasing pressure increases proton permeability in the squalene-free LUVs, at least at the temperature, presumably about 10°C , at which the pressure experiments are done. The authors have commented on this in the two paragraphs (lines 308-338) following the description of Figure 5 in the text. The authors do list some possible

mechanisms for proton conductance across lipid bilayers. These all seem to depend, to some extent, on penetration of water or chains of water (wires) but the laurdan experiments indicate that pressure reduces water penetration. In the two paragraphs following the discussion the authors do comment on the relative insensitivity of bilayer thickness to pressure in the absence of squalene. This likely does imply some frustration. Since the laurdan observations indicate that pressure reduces water penetration in the environments sampled by most of the bilayer lipids, is it possible that pressure promotes the emergence of proton permeation sites that are not uniformly distributed over the bilayer surface but, rather, are clustered at the edges of highly ordered regions (large area to perimeter ratio) with very low water penetration? If so, such "defects" would have to be small enough to not contribute significantly to CF efflux. Regardless of the mechanism by which pressure induces proton transport in the absence of squalene, the authors note that squalene at the bilayer midplane blocks proton permeation, as might be expected.

The manuscript goes on to relate these observations to the role that apolar intercalants might play in maintaining functionality of diether lipid bilayers at high temperature and high pressure. The authors have demonstrated that diether lipid bilayers containing squalene at the midplane are able to maintain the compressibility, fluidity, and proton impermeability required for membrane function. This is a significant finding. The authors do a good job of putting these findings into the context of knowledge about how organisms adapt to extreme conditions. Overall, I think that this is an interesting, significant, and very well executed contribution.

I did note some additional minor points regarding the manuscript.

On page 9 (lines 202 to 204), it would be helpful to more fully hyphenate "large-negative-curvature" and "large-positive-curvature".

On page 11 (line 257) the "(43)" looks like it might have been a reference using a different citation format.

On page 12, lines 281 and 282, some punctuation is missing from the citation and there is a spurious "#27".

On page 13, line 303, I would suggest adding something like "With squalene present, " in front of "increasing pressure" to better cue the reader to the intent of the statement.

In panel D of Figure 2, the lines that connect the lipid chain region in the left bilayer to the lipid chain region on the right bilayer also appear to line up with DC. It is pretty clear from the table below the bilayer drawings that DC extends to the middle of the squalene layer but perhaps there is a way to indicate that more clearly, perhaps by showing DC for the intercalate bilayer to the right of the picture.

As already noted, it would be helpful, in the caption of Figure 5, to include the temperature at which the variable pressure experiments were done.

In the caption to Figure 6, I think it should be "apolar" rather than "non-apolar".

In the methods, the equation for the density seems to have a spurious "square" in it. My guess is that the stuff following the square should have gone into the square in the equation editor.

On line 610, why is "laurdan" double underlined?

As already noted, the parts of the methods describing the LUV experiments should indicate the temperature at which the pressure scans were carried out.

Reviewer #3 (Remarks to the Author):

In their manuscript, Salvador Castell and co-workers study i) the location and molecular orientation of one apolar intercalant (squalene) in synthetic lipid bilayers composed of two archaeal phospholipids (DoPhPC and DoPhPE) and ii) the effect of temperature and pressure on the synthetic membrane-like structures in the presence and absence of the intercalant. The authors appear to convincingly show that squalene intercalates between the two phospholipid monolayers in parallel to them. The presence of this apolar intercalant appears to have a marked effect, affecting curvature and reducing permeability to protons but increasing permeability to water. The latter effects are particularly noticeable as temperature and pressure increase in comparison with archaeal lipid bilayers devoid of the apolar intercalant. Therefore, such membrane architecture might constitute an adaptation of archaea endowed with bilayer lipid membranes to high temperature.

The experiments seem carefully and rigorously conducted and the observations in the synthetic membranous structures, solid. However, the data derived from these interesting experiments are overinterpreted and excessively stretched to make conclusions that are not necessarily supported by these particular data. The manuscript needs to be toned down and conclusions need to be adapted to the experimental observations.

Specific concerns:

- My major concern resides in the fact that the authors make observations about the properties of synthetic lipid bilayers, but not of natural membranes. However, they make conclusions about archaeal membranes, which are certainly more complex, including proteins, etc. and for which a similar architecture to that observed in synthetic structures remains to be demonstrated. I understand the advantages of using synthetic membrane-like bilayers. Indeed, this is the most direct way to show the effects of the apolar intercalant on the archaeal-type phospholipid bilayer under controlled conditions. But this allows at best to formulate hypotheses about what might occur in natural archaeal membranes. Therefore, in the absence of data on true archaeal membranes, statements about archaeal membrane properties (which are pervasive in the manuscript, starting by the title "A novel membrane ultrastructure explains the cell adaptation...") should be removed and placed under the status of hypothesis in the discussion section.

- Likewise, only one intercalant is tested (squalene), yet the authors conclude that apolar isoprenoid intercalants, in general, have the same effect as observed here. Again, this generalization is not allowed by the observation of only one molecule's effects and should be placed as a (perhaps likely) hypothesis in the discussion.

- The observations about phospholipid layer curvature changes are interesting, as are the potential implications for archaeal membrane remodelling. However, this is again hypothetical at best as there is no evidence that those membrane remodelling processes in archaea are actually based on apolar isoprenoids intercalating in the midplane of bilayers. The idea is tempting, but those processes might be simply achieved in cells by the presence of membrane-bending proteins.

- Page 3, lines 62-63. Only one particular strain of *Methanopyrus kandleri* (ko 116) has been shown to

grow up to 116°C at atmospheric pressure, 122°C under higher pressures. This upper boundary limit appears to be conditional and possibly needs validation.

Answers to reviewers' comments:

We would like to express our thanks for the comments and encouragements of the three reviewers who helped greatly improve the presentation of our work.

We hope to have addressed all comments appropriately in the current revised manuscript.

Philippe Oger

Reviewer #1 (Remarks to the Author):

In the manuscript "A novel membrane ultrastructure explains the cell adaptation to extreme conditions in Archaea" by Castell et al., the authors report the effect of the insertion of squalene into an archaeal lipid bilayer analogue. First they address the insertion of the squalene in the bilayer. Later on, they study the effect of the squalene insertion on the structure and function of the artificial membrane by analyzing for example membrane curvature as well as water, solute and proton permeability in response to temperature and pressure change.

The studies were performed by using small-angle X-ray scattering (SAXS), neutron diffraction experiments as well as spectroscopic experiments with unilamellar lipid vesicles using fluorescent dyes.

With their artificial system they show the insertion of the squalene in the midplane of the bilayer increase the permeability of water and reduce the permeability of protons under extreme conditions.

In addition, apolar isoprenoids induce a negative curvature in the membrane.

The authors represent a thorough and interesting analysis that fits nicely in the current discussion of the origin of life. However, in order to be of relevance for a broader public the authors should consider to give more details about the current understanding of the archaeal membrane composition and clearly state the differences to their artificial system (for further details see below).

L20 (Abstract) In the abstract the performed experiments and major findings of this study are not clearly presented. Maybe it would be helpful to include their experiential approach here.

The abstract has been largely rewritten for clarity. We hope that this is now more effective.

L44 (Introduction). Here the authors have to present a more thorough presentation about the current knowledge of the archaeal membrane in order to allow the comparison to their artificial model. Thus, it has to be stated that the polar head group is formed by glycerol 1-phosphate (not glycerol-3-phosphate). What are the phosphate-bound substituents in Archaea and of interest for this study of *Methanopyrus kandleri*?

These precision have been added in the section describing in detail the experimental system (lines 140-156).

If the reviewer refers to the nature of the polar headgroups in these two organisms, they are essentially unknown (see a more detailed explanation below, in response to comment on line 113).

In this regard it might be also interesting that membrane fluidity in Archaea is regulated by the introduction of heterocycles whereas unsaturated fatty acids are found in Bacteria and Eukaryotes. Finally, as in Bacteria and Eukaryotes numerous proteins are found in the membrane.

We agree that a little more explanation would be helpful to the reader and to clarify that the work done here is performed on a very simplified membrane analogue. To do so, we have largely restructured the introduction. Modifications are in red throughout the text.

L63 The authors might want to give the growth range and growth optimum.

Aeropyrum pernix growth range: 70 °C – 100 °C (doi: 10.1099/00207713-46-4-1070)

Aeropyrum pernix growth optimum: 90 °C – 95 °C (doi: 10.1099/00207713-46-4-1070)

Methanopyrus kandleri growth range: 85 °C – 116 °C at 0.4 MPa, 122°C at 20 MPa (doi: 10.1073/pnas.0712334105)

Methanopyrus kandleri growth optimum: 100 °C at 0.4 MPa (doi: 10.1073/pnas.0712334105)

We appreciate the suggestion. We have added the optimal growth temperature in the main text.

L84 This passage is rather speculative. It seems to be more appropriate for a final discussion and in my opinion the authors need to be very clear and careful what conclusions they can retrieve from their artificial archaeal membrane model.

As suggested by the reviewer, we have moved part of this message to the discussion section. However, we think that it also illustrates the biological importance that this novel membrane structure may have and thus, taking all the due precautions, we have decided to maintain some of these notions in the introductory section.

L91/92 The sentence should be rephrased to: Last, the architecture also implies the existence of membrane domains of divergent compositions "in the archaeal membrane".

Thank you, we have rephrased it, now it reads “Last, this architecture also implies the existence of membrane domains of divergent compositions in the archaeal membrane...”

L113 To what extent does the model represent the bilayer structure of Thermococcus barophilus? The authors might want to include why they decide on the chosen archaeal-like lipids DoPhPC and DoPhPE. Was it due to the commercial availability?

The information about headgroups of archaeal lipids remains scarce, and it is even more so for *Thermococcus barophilus*. This species was originally described to produce only diether phospholipids, with a phosphatidylinositol moiety. Two different studies, have later shown that diether lipids represent only ca. 50% of the phospholipids. The quality of the polar headgroups is yet to be determined. In our current unpublished results, we have thus far identified more than 14 different core lipid structures and 13 different types of polar headgroups, for a total of 84 different intact polar lipids (Tourte et al., unpublished). Unfortunately, these analyses also show that we have been able to analyze only a small fraction of the lipid diversity in this species, amounting to less than 10% of the total lipid content, and that we might be missing some of the major lipids.

If we enlarge the answer to the whole Archaea domain, a large variety of lipid headgroups have been identified, amongst which, the PC and PE used in this study, but

also PS, PG, PI and their derivatives.

Thus, we have chosen to use PC and PE as a first approximation due to its commercial availability (and therefore their high purity). We expect it to be representative of the behavior of at least some archaeal lipids, since PC and PE lipids have been reported in many Archaea to date.

We have added this information in the revised paragraph on the localization of apolar polyisoprenoid (lines 140-156).

L128 The lycopane structure might be shown in comparison to the squalene structure in Fig. 1.

Thank you for your suggestion, we have added it.

L128 The Fig 2B needs to be explained in more detail. What is shown here?

Figure 2B is a sketch of a top view looking down on the lipid bilayer, the top view of lipid headgroups are represented in red and the CH₃ groups of an apolar polyisoprenoid as black spheres. Our intention was to show another view of the squalene in the lipid bilayer to manifest how only 1 mol% of squalene have this much effect on the membrane.

We have clarified the legend, now it reads: “Sketch of a top view of the interaction of the apolar lipid squalene (methyl groups represented as black spheres) in the midplane of the reconstructed archaeal-like lipid bilayer membrane (lipid isoprenoid chain extremities are represented in red).”

L137 From the results shown in Fig 2C can it be excluded that the squalene intercalates partially into both monolayers and thus connects them to a monolayer. I am not familiar with the respective methods, but this is what I would expect from hydrophobic molecules and the NSLD observed might fit approximately to the size of the squalene.

We appreciate the comment and we understand the interrogation. The fact that the NSLD shows a gaussian distribution at $D=0 \text{ \AA}$ indicates that, although with a certain degree of mobility, squalene is mostly located in the bilayer midplane. If squalene had been located parallel to the lipid hydrophobic chains, the NSLD would have presented a broader increase further from the bilayer midplane. Moreover, the detailed analysis of the neutron diffraction results manifests that the presence of squalene increases the thickness of the membrane hydrophobic region (D_c), effect that can only be explained if squalene is mostly located in the bilayer midplane, perpendicular to the axis normal to the membrane plane. Finally, the fact that squalene induces a higher negative curvature to the membrane indicates the squalene position that we proposed here is correct: squalene releases the membrane packing tension by filling the bilayer midplane voids in these higher curved structures.

Also, the authors should discuss if the preparation method might result in artifacts.

The preparation method and the technique used has been extensively used to localize other molecules in lipid bilayers (e.g. cholesterol) and it has been shown to be one of the most reliable and accurate technique for this purpose.

L171 For me the phase discussion it is a bit hard to follow. May be the authors can explain initially what kind of different phases and transitions can be observed using their experimental approach.

We have added the following additional explanations: “Lipids spontaneously organize in different phases, the most well-known being the lamellar phase, e.g. the lipid phase of the plasmic membrane. But depending on the local environment, they can also organize into the so called non-lamellar phases, such as the cubic or hexagonal phase (See left top panel in figure 3). Our results reveal that non-lamellar phases (inverted bicontinuous cubic and inverted hexagonal) are only formed in presence of squalane, and that these phases are also promoted by high temperatures.”

L256 The authors might want to mention that they use unilamellar liposomes for the spectroscopic permeability assays.

Thank you, we added this information in the main text.

L282 As far as I know water is typically transported via membrane proteins (water channels, aquaporines) in living cells. These have been also reported for Archaea e.g. <https://www.pnas.org/content/102/52/18932>.

We fully agree with the comment. It was ill presented in our draft. The aim of this experiments is not to test or mimic the water permeation of the natural membrane, which indeed occurs mostly via pores. However, the impermeability of the lipid system in itself is important since only it can ensure that water fluxes through the pores can be regulated. We have rewritten this passage to read as follows "This is used to mimic the transfer of water through the lipid layers. Thus, it does test the intrinsic impermeability of the lipid layers, which needs to be high enough in order for the cell to control its inward and outward water flows, which occurs in cells by the mediation of transmembrane proteic water channels.”

L376 In the discussion the possible roles of proteins that can be not addressed using the archaeal membrane model needs to be considered. Besides aquaporines for water transport also proteins are discussed to be involved in vesicle formation for example the ESCRT system in Crenarchaeota, FtsZ-based system in Euryarchaea <https://www.nature.com/articles/nrmicro2406>.

We expand the possible roles of proteins in the discussion. Now it reads: “In this study, we have used a model of an archaeal lipid bilayer and thus, features of archaeal membranes (e.g. variety of lipids, presence of membrane proteins) are not represented.”

L392 There is quite some insight into the role of vesicles in Archaea.

We have added another example of the role of vesicles in Archaea.

Now it reads: “Although it is not completely clear what role these vesicles play in the archaea life cycle, they are proposed to be involved in genetic exchange and used to complicate the growth of competing species and(Gaudin et al., 2013; Deatherage, et al. 2012).”

L403 In the final discussion it should be clarified that this is only a model and that typical features for the archaeal membrane (see discussion above) are missing.

We agree that it is a very simplified model, we put more emphasis on this in the current revision. Now it reads: “In this study, we could only use a simplified analog of an archaeal lipid bilayer and thus, common features of natural membranes (e.g. variety of lipids, presence of membrane proteins) and specific ones of archaeal membranes (glycerol stereochemistry) have not been tested. In addition, the presence of apolar polyisoprenoids in archaeal membranes still needs to be explored in depth, since it has not been searched for nor quantified in most of them (Review IJMS). Taking this into

account, here we made a first experimental validation of a molecular model that may help to understand alternate adaptive routes to membrane adaptation to high temperature and support the structural and functional stability of the membrane of the extremophilic Archaea lacking the ability to synthesize monolayer-forming membrane lipids. Most importantly, the membrane model architecture explored here has possible implications in the way we understand the evolution of membrane adaptation to extreme conditions, especially with regard to the role of the bipolar lipids.”

L465 Fig. 2 D The model shows glycerol 1-phosphate (see fig. 1) in the experiment glycerol 3-phosphate is used a head group.

Thank you for noting this out, we have changed the model from figure 2D and also figure 5, which had the same issue.

Reviewer #2 (Remarks to the Author):

This paper describes an experimental study of model membranes composed of diether phospholipids with or without the apolar isoprenoid squalene at the midplane. Interest in this membrane architecture is motivated by the relative abundances of diether versus tetraether lipids in some extremophile Archaea membranes. The rationale for employing synthetic lipids in this study is clearly presented and convincing. The authors employ neutron scattering, small-angle X-ray scattering, and permeability assays to characterize these model membranes at different temperatures and pressures and to assess how the presence of the apolar intercalant might contribute to the adaptation of organisms to extreme conditions.

The reported observations are of high quality and support the interpretations offered by the authors. I think that this is a significant and novel contribution. It provides some very interesting insights into a membrane architecture that appears to enable some organisms to maintain functional membranes over wide ranges of temperature and pressure. It also poses some interesting questions about aspects of this class of membrane, particularly its response to applied pressure.

The comparison of neutron scattering length densities for the diether phospholipid bilayers containing either hydrogenated or deuterated squalene provide an unambiguous demonstration of squalene's location at the bilayer midplane. Small-angle X-ray scattering was used to characterize phases observed over a range of temperatures at ambient pressure and at 1000 bar for bilayers without and with squalene. The insensitivity of d-spacing to pressure in the absence of squalene as shown in Figure 3 and in Figure S1 is striking. I would have been interested in a bit more detail about the two lamellar phases found to coexist in the absence of squalene.

There is very little we can say about these two lamellar phases with the data that we have obtained here. The closeness of their d-spacing suggest they are very similar in structure of lipid composition, but there is no real hint at whether this is a phase separation due to a partitioning of the lipids in different parts of the membrane (our favorite hypothesis), or whether this is only a physical modification like a tilting of the membrane lipids.

We have added the following sentence to the revised draft to explain in some length the observation on the lamellar phases "The small difference observed in their d-spacing suggests that they are quite similar in structure. Several reasons could lead to this, such as a variation of the lipid composition due to lipid partitioning affecting the polar or apolar lipids, or both, or a tilting of the lipids in the membrane. The data acquired here does not allow really to extrapolate at length on these coexisting lamellar phases at this point."

The presence of squalene also promoted the coexistence of a cubic phase with either lamellar or hexagonal phases and increased the sensitivity of the lamellar and hexagonal structures to pressure. The authors reasonably attributed the accommodation of non-lamellar organization to structural flexibility provided by the presence of squalene at the midplane.

The authors used laurdan generalized polarization (GP) to assess chain order (as a proxy for water penetration into the model membranes), carboxyfluorescein (CF)

efflux to assess permeability to larger molecules (and as a proxy for permeability to water), and pyranine fluorescence to assess permeability of the membranes to protons, presumably as H_3O^+ . These assays were done as functions of temperature and pressure in LUV bilayers without and with squalene. I was unable to find a statement about the temperature at which the pressure dependent LUV studies were carried out but based on comparison of the ambient pressure results from panels C, F, and I of Figure 5 with the corresponding temperature dependent observations in panels B, E, and H, I am guessing that the pressure experiments were done at about 10°C. It would be helpful to have some comment about the rationale for this choice of temperature for the LUV experiments and whether the pressure dependent behavior might have been different at higher temperature since high temperature and high pressure might be an interesting condition to probe for this system. If there was an experimental constraint on temperature for the high pressure LUV experiments, a comment would be appropriate.

All HP LUV experiments were done at 20°C. This is now said in the revised draft. Indeed there was a technical constraint of the instrument we used for this series of measure, which only allowed to use either pressure or temperature regulation. Thus, the very meaningful joint P and T experiments could not be performed.

The laurdan GP observations suggest that squalene has a small ordering effect on the bilayer. Squalene does not substantially alter the sensitivity of the laurdan GP to changes in temperature or pressure. This is probably not too surprising since laurdan is not sampling the membrane environment close to the midplane. The interpretation of the laurdan GP observations seems to be based on existing results for laurdan in bilayers of diester lipids. The authors might want to consider how differences between hydration of the diether and diester lipids might be reflected by laurdan GP. There is some discussion of the effects of ether versus ester linkage on laurdan GP in a paper by Perez et al. [Perez et al. (2021) Effect of cholesterol on the hydration properties of ester and ether lipid membrane interphases. *Biochim. Biophys. Acta* 1863, 183489]. The Perez et al. paper appeared online only recently, presumably after the current manuscript was submitted, but maybe there is some earlier literature on laurdan GP in diether lipids that might be pertinent. I don't think that differences in laurdan GP due to the ether linkages would alter the main findings that squalene has only a small effect on order and that order increases with pressure but some additional context would be helpful.

Thank you for mentioning this very interesting study about the difference in GP of laurdan depending on the nature of the chemical bound of the acyl chains to the glycerol moiety. We have introduced some context to the revised manuscript in order to make these points somewhat clearer.

Panel E of Figure 5 shows that squalene increases the permeability of the vesicle membranes to CF and thus, presumably, to molecules like water. The authors reasonably attribute the squalene-induced increase in CF efflux with increasing temperature to phase coexistence and domain boundaries. The observation, in panel F of Figure 5, that squalene does not change the sensitivity of CF influx to pressure is not too surprising. If this experiment was done at a temperature corresponding to the lowest temperature in panel E of Figure 5, that would be a temperature at which CF efflux is already low and largely independent of the presence of squalene. Increasing the pressure at this temperature would likely

keep the efflux at a low level (possibly reflected by the large error bars in panel F).

Indeed, the experiments were performed at a temperature close to the lowest (20°C) at which the impact of squalene is low, and possibly the increase in pressure reduces the formation of phase boundaries and such defects which promote CF leakage. The large error bars in panel F are in great part due to the HP apparatus and the difficulties to acquire fluorescence in these conditions. To connect this comment and one of the ones above, it would have been very interesting to test to what extent the HP could counteract the impact of increasing temperature on CF leakage.

The effects of temperature and pressure on the permeability to protons in the absence and presence of squalene, shown in panels G, H, and I of Figure 5 is quite interesting. While panel G does illustrate the connection between increasing permeability to protons and decreasing intensity, it might not hurt to explicitly mention that relationship in the text or in the figure caption. The authors note that squalene has no effect on proton permeability at ambient pressure which is indeed surprising. It is particularly interesting, though, that increasing pressure increases proton permeability in the squalene-free LUVs, at least at the temperature, presumably about 10°C, at which the pressure experiments are done. The authors have commented on this in the two paragraphs (lines 308-338) following the description of Figure 5 in the text. The authors do list some possible mechanisms for proton conductance across lipid bilayers. These all seem to depend, to some extent, on penetration of water or chains of water (wires) but the laurdan experiments indicate that pressure reduces water penetration. In the two paragraphs following the discussion the authors do comment on the relative insensitivity of bilayer thickness to pressure in the absence of squalene. This likely does imply some frustration. Since the laurdan observations indicate that pressure reduces water penetration in the environments sampled by most of the bilayer lipids, is it possible that pressure promotes the emergence of proton permeation sites that are not uniformly distributed over the bilayer surface but, rather, are clustered at the edges of highly ordered regions (large area to perimeter ratio) with very low water penetration? If so, such "defects"; would have to be small enough to not contribute significantly to CF efflux. Regardless of the mechanism by which pressure induces proton transport in the absence of squalene, the authors note that squalene at the bilayer midplane blocks proton permeation, as might be expected.

The manuscript goes on to relate these observations to the role that apolar intercalants might play in maintaining functionality of diether lipid bilayers at high temperature and high pressure. The authors have demonstrated that diether lipid bilayers containing squalene at the midplane are able to maintain the compressibility, fluidity, and proton impermeability required for membrane function. This is a significant finding. The authors do a good job of putting these findings into the context of knowledge about how organisms adapt to extreme conditions. Overall, I think that this is an interesting, significant, and very well executed contribution.

Thanks you very much for all the positive comments.

I did note some additional minor points regarding the manuscript.

On page 9 (lines 202 to 204), it would be helpful to more fully hyphenate "large-

negative-curvature" and "large-positive-curvature".
Modified as suggested.

On page 11 (line 257) the "(43)"; looks like it might have been a reference using a different citation format.

On page 12, lines 281 and 282, some punctuation is missing from the citation and there is a spurious "#27".

Thank you for noting this out, we have fixed these citations.

On page 13, line 303, I would suggest adding something like "With squalene present, " in front of "increasing pressure"; to better cue the reader to the intent of the statement.

Thank you for your suggestion, we have added it.

In panel D of Figure 2, the lines that connect the lipid chain region in the left bilayer to the lipid chain region on the right bilayer also appear to line up with DC. It is pretty clear from the table below the bilayer drawings that DC extends to the middle of the squalene layer but perhaps there is a way to indicate that more clearly, perhaps by showing DC for the intercalate bilayer to the right of the picture.

Thank you for the suggestion, we think that indeed it may help to understand the results. We have added it into Figure 2.

As already noted, it would be helpful, in the caption of Figure 5, to include the temperature at which the variable pressure experiments were done.

Thank you for the suggestion, we have added it to the figure.

In the caption to Figure 6, I think it should be "apolar"; rather than "non-apolar".

Fixed.

In the methods, the equation for the density seems to have a spurious "square"; in it. My guess is that the stuff following the square should have gone into the square in the equation editor.

Absolutely right, thank you for noting it out.

On line 610, why is "laurdan"; double underlined?

It was an typo, we have fixed it.

As already noted, the parts of the methods describing the LUV experiments should indicate the temperature at which the pressure scans were carried out.

We have added this information (20°C).

Reviewer #3 (Remarks to the Author):

In their manuscript, Salvador Castell and co-workers study i) the location and molecular orientation of one apolar intercalant (squalene) in synthetic lipid bilayers composed of two archaeal phospholipids (DoPhPC and DoPhPE) and ii) the effect of temperature and pressure on the synthetic membrane-like structures in the presence and absence of the intercalant. The authors appear to convincingly show that squalene intercalates between the two phospholipid monolayers in parallel to them. The presence of this apolar intercalant appears to have a marked effect, affecting curvature and reducing permeability to protons but increasing permeability to water. The latter effects are particularly noticeable as temperature and pressure increase in comparison with archaeal lipid bilayers devoid of the apolar intercalant. Therefore, such membrane architecture might constitute an adaptation of archaea endowed with bilayer lipid membranes to high temperature.

The experiments seem carefully and rigorously conducted and the observations in the synthetic membranous structures, solid. However, the data derived from these interesting experiments are over-interpreted and excessively stretched to make conclusions that are not necessarily supported by these particular data. The manuscript needs to be toned down and conclusions need to be adapted to the experimental observations.

Specific concerns:

- My major concern resides in the fact that the authors make observations about the properties of synthetic lipid bilayers, but not of natural membranes. However, they make conclusions about archaeal membranes, which are certainly more complex, including proteins, etc. and for which a similar architecture to that observed in synthetic structures remains to be demonstrated. I understand the advantages of using synthetic membrane-like bilayers. Indeed, this is the most direct way to show the effects of the apolar intercalant on the archaeal-type phospholipid bilayer under controlled conditions. But this allows at best to formulate hypotheses about what might occur in natural archaeal membranes. Therefore, in the absence of data on true archaeal membranes, statements about archaeal membrane properties (which are pervasive in the manuscript, starting by the title "A novel membrane ultrastructure explains the cell adaptation") should be removed and placed under the status of hypothesis in the discussion section.

We understand the worries of the reviewer and consequently, we have put more emphasis on bringing out that this is only a simplified artificial membrane model and we only discussed several plausible implications that this architecture would have in native cells.

To this avail, several paragraphs of this revised manuscript have been rearranged to replace our conclusions in the scope of the simplified model, and to make very clear when the results from this model are extrapolated to the natural membrane. All modifications appear in red in the current revision.

- Likewise, only one intercalant is tested (squalene), yet the authors conclude that apolar isoprenoid intercalants, in general, have the same effect as observed here. Again, this generalization is not allowed by the observation of only one molecule's effects and should be placed as a (perhaps likely) hypothesis in the discussion.

We could indeed only fully assay one apolar molecule in depth using all these

techniques. The reviewer is correct that we may have over extrapolated our results. We have reviewed this matter and specified that we refer to only squalane.

- The observations about phospholipid layer curvature changes are interesting, as are the potential implications for archaeal membrane remodeling. However, this is again hypothetical at best as there is no evidence that those membrane remodeling processes in archaea are actually based on apolar isoprenoids intercalating in the midplane of bilayers. The idea is tempting, but those processes might be simply achieved in cells by the presence of membrane-bending proteins.

This is a very interesting consideration and a long lasting interrogation of the community. How/to what extent does the membrane protein interact with its surrounding lipid environment? Does any hydrophobic mismatch between lipid and protein lead to distortion of the lipid bilayer, to distortion of the protein, or to distortion of both? How are the imperfections created in the membrane by the insertion of the protein in the membrane coped with? What lipids are compatible with the proper function of membrane proteins? How the phase of the lipid (liquid crystalline, gel or hexagonal HII) affects the function of a membrane protein? There is a large body of literature on this issues.

We agree that membrane-bending proteins are essential on some cellular processes, but we think that the literature does not support the role of these proteins as the only players in the insertion of membrane/membrane associated proteins in the membrane. Indeed, the presence of these non-lamellar forming lipids in native membranes has been extensively reported and their presence has been proven to decrease the bending membrane energy, hence allowing high curvature in the membrane. It is thus reasonable to assume that they partake in helping bend the membrane.

It is quite difficult to determine with precision which lipid of the membrane interacts with which part of a protein with precision. In fact, most methods with sufficient resolution such as Xray crystallography imply strong modifications of the protein local environment and thus even when lipids were found associated with the protein crystals, there is always some uncertainty as to whether these were lipids normally in interaction with the protein, and when so, whether they were in their natural position.

If we consider membrane proteins that have been studied to some molecular details in their natural addition, and the particular example of Archaea, to my knowledge there is only one system that has been reported to date, e.g. bacteriorhodopsin from the hyperhalophilic archaeon *Halobacterium salinarum*. In this example, it has been shown that the membrane is composed of ca. 30% of protein, which implies a great necessity for membrane curvature. In this context, up to 20% of the lipids is squalene, an unsaturated derivative of squalane. Most of the other lipids have very bulky headgroups, which implies that they will allow for high positive membrane curvatures. Unfortunately, the protein-lipid interaction has not yet been described at the molecular scale. However, given the requirement for highly curved membranes, it is expected that numerous structural defects will be generated in the membrane. Thus, given the similarity of the apolar molecules, it is largely conceivable that squalene fills the voids left in the membrane by the presence of the large proportion of high curvature lipids needed to fit the extremely large proportion of protein of the purple membrane as well as help relieve lipid frustration as proposed here.

Nonetheless, we agree that this is at this point speculative, and we have reworked the manuscript to tone down this part and mentioned it as an extrapolation. But due to the biological importance of it, we felt that it was important to maintain this part of the

discussion to raise an interest in more detailed studies of protein-lipid interactions at the molecular scale in natural systems.

- Page 3, lines 62-63. Only one particular strain of Methanopyrus kandleri (ko 116) has been shown to grow up to 116°C at atmospheric pressure, 122°C under higher pressures. This upper boundary limit appears to be conditional and possibly needs validation.

To date there has been no study disproving this limit, so we only can report it as published. However, we have amended the draft to provide the growth information for atmospheric pressure as well. We have specified these values in the main text. Now it reads: "... growth of 122°C at 20 MPa (T_{opt} = 100°C at atmospheric pressure) (Takai et al., 2008)"

REVIEWERS' COMMENTS:

Reviewer #1 (Remarks to the Author):

I had another careful look at the rebuttal letter as well as the revised manuscript. All points were addressed adequately to my full satisfaction. Therefore I have no further comments.

Reviewer #2 (Remarks to the Author):

See attached file

Reviewer #3 (Remarks to the Author):

Overall, the manuscript has been considerably improved, although there are some typos and/or small grammatical errors that might be corrected.

I still have a few minor comments:

- Lines 66-67, "The recent discovery of monolayer, ether lipids in the membranes of the most hyperthermophilic bacteria gave further support to the view that tetra-ether lipids were the adaptation route to life at high temperatures." The authors do not provide a reference here, but they should. Ether-linked lipids in hyperthermophilic bacteria have been known for over 20 years now, although some recent papers just ignore that information. It would be good to provide appropriate original citations.
- 122°C as an upper temperature for *Methanopyrus kandlerii*. The explanation of the authors would be adequate in an ideal world. However, it is not because something is published that it is necessarily true and does not need validation. Life at 250°C was reported in one top-journal, never to be reproduced! 122°C may need validation as well.
- That membrane-bending proteins have an effect on membrane remodelling is undeniable. That this is linked to particular membrane phospholipid rearrangements is possible. The authors should recognize this. A clear example is that of membrane-remodelling proteins in Asgard archaea. The particular morphology of *Prometheoarchaeum syntrophicum* with extensive membrane protrusions is most likely largely caused by membrane-remodelling ESCRT-like proteins.

Point per point answer to reviewers comments:

We would like to take the opportunity to again thank the reviewers to help us improve the quality of the presentation of our work in the present manuscript.

Reviewer #1 (Remarks to the Author):

I had another careful look at the rebuttal letter as well as the revised manuscript. All points were addressed adequately to my full satisfaction. Therefore I have **no further comments**.

Reviewer #2 (Remarks to the Author):

- The authors have mostly addressed the concerns raised in my initial review of this manuscript. There is however one issue from that review that I feel still needs some attention. I noted that the variable pressure experiments seemed to have been carried out only at room temperature. The authors confirmed this and have now indicated the temperature for the variable pressure experiments and indicated that temperature on the axes of panels C, F, and I of Figure 5. It is unfortunate that it was not possible to explore the high temperature, high pressure conditions most pertinent to extremophile environments but I don't think that negates the significance of the interesting findings reported in this work. I did suggest that some comment was needed and I would still say that this is still the case. I think that there needs to be a clear and explicit statement that technical constraints precluded exploration of the effect of simultaneous high pressure and high temperature conditions and there needs to be some comment on how doing the pressure experiments at ambient pressure might have affected the observations and the capacity to draw conclusions about the response of the model system to simultaneous high pressure, high temperature conditions. This is an issue because, as the authors do note, the ordering effects of high pressure can mitigate effects of high temperature. I think it is up to the authors to determine where it would be best to comment on this issue but near the end of the discussion might be an appropriate place to do so.

We have modified the discussion to emphasize the need for further work on combined P and T experiment to explore their overlapping and antagonistic impacts on membrane properties, especially in the light that the role of squalane might be to regulate these antagonistic effects. This paragraph now reads: "Our results confirm and expand on the hypotheses discussing novel membrane architectures as proposed by Cario and colleagues (Cario et al., 2015). Importantly, we provide an experimental demonstration that in this membrane architecture, the lipid bilayer stability and functionality are shifted to higher temperatures under high pressure, which supports the view that apolar lipids may constitute one of the adaptative routes to high-temperature tolerance in that could exist in archaeal hyperthermophiles. In the future, it

will be important to explore the relative contribution of the different polar and apolar lipids on membrane parameters as a function of combined hydrostatic pressure and temperature, since increasing hydrostatic pressure and temperature have antagonistic effect on molecular systems, including lipids. Indeed, for instance increasing temperature will increase molecular motion, while increasing hydrostatic pressure will decrease it. Thus, the presence of the apolar lipid might be play an important role in the fine regulation of membrane properties under combined high pressure and temperature."

- In my initial review, I indicated my assessment that the reported observations are of high quality and that this is a significant and novel contribution. I still feel that it provides some very interesting insights into and raises interesting questions about how membrane architecture might contribute to the capacity of organisms to respond to extreme conditions. I have noted some possible issues with wording in some of the revised sections and I have identified some issues with wording that might not have been as apparent when I read the initial version of the manuscript. I will also note that, in my previous review, the autocorrect function in my word processing software seems to have consistently substituted "squalene" when I typed "squalane". I think I have fixed that this time and I apologize for any confusion it may have caused in my earlier review.

Several corrections were made throughout the draft to correct wording and typos. We hope we have adressed all possible issues regarding this point.

- On Line 59, "can" should probably be replaced by "may" since the idea that compact, impermeable membranes might explain the tolerance of Archaea to extremes seems to be a suggestion based on the observed compositions of their membranes. Overall, the revised Introduction now does a better job of providing the context and rationale for this work.

Replaced.

- In the revised section starting at line 138, there is a statement that starts "One of the drawbacks of this approach is that to date no natural or pure archaeal lipid is available commercially. "I don't think that this sentence or the following sentences properly convey the intended meaning. The lack of commercially available archaeal lipid is a constraint. The drawback is that the system then has to be modeled using synthetic lipids with glycerol moieties having bacterial stereochemistry. This could be addressed by a slight rewording.

We thank the reviewer for this very pertinent remark. The drawback here is more on the artificial and simplistic nature of the reconstructed membrane rather. The lack of phospholipids in the archaeal stereochemistry is more of a further constraint. Thus, we have modified this sentence to the following to better state it. "One of the constraint of this approach is that there are no archaeal lipids available commercially to date, and no phospholipids harboring all

the archaeal lipids features, especially the *sn*-glycerol-1 phosphate, e.g. the archaeal stereochemistry of the glycerol moiety."

- With the insertion of the revised section between lines 138 and 154, there is now a rather awkward transition from the description of the model system to a description of the neutron diffraction results. To properly highlight the neutron results, perhaps there should be a new paragraph started at "Protonated (H-squalane) and perdeuterated (D-squalane) pure squalane....".

Modified accordingly.

- On line 170, the "d-spacing" should probably be identified as the periodicity of the bilayer structure rather than the more ambiguous wording "distance between two bilayers including the interlamellar water thickness".

We have modified this sentence for clarity as suggested. It now reads: "The d-spacing represents the periodicity of the bilayer structure, e.g. the distance between two bilayers in the stacks, including the interlamellar water thickness."

- Lines 199 to 204 contain new text presumably added in response to a reviewer comment. I don't think that it is particularly well located. I would suggest that sentence "Lipids spontaneously organize..." and the next sentence, which starts "But depending on the local environment..." be moved to the beginning of this paragraph. That would introduce the idea of lipid phases and the paragraph would then continue describing how SAXS was used to explore the possibility of phase transitions.

Modified accordingly.

- The sentence that starts on line 202 with the words "Our results reveal..." is out of place and the statement that non-lamellar phases form only in the presence of squalane needs to be qualified to indicate that the dependence on squalane is specific to the model membranes studied here. I would suggest moving this sentence, with the appropriate qualification, to the beginning of the next paragraph.

Modified accordingly. We added the qualification "in this model membrane system" to be more precise on the specificity of this observation to our model system.

- The revised section on lines 206 and 207 describes how the lamellar phase was identified from the SAXS scattering intensity versus scattering vector plots in Figure 3. The wording "as can be seen in the periodicity of the Bragg peaks along a sequence: 1, 2, 3, etc." probably does not convey the intended meaning. It is the sequence of scattering vectors corresponding to Bragg peaks that is periodic and the sequence is the ratio of those scattering vectors. An improved wording might be "...as can be seen in the ratios (1:2:3:..) of the scattering vectors at which Bragg peaks are observed". The wording in the figure caption should be corrected too. The wording in the Methods section is better but I think that it would be preferable to indicate ratios by

using colons to separate numbers in the sequences (which means that the colon before each sequence would need to be removed).

Modified as suggested.

- In the next sentence, starting on line 207, the reference to “a known specific characteristic..” is ambiguous. The sentence could be read as meaning that the absence of the transition is characteristic or the transition itself is characteristic. This could be improved to clarify that it is the absence of a transition that is characteristic by a slight rewording to something like “The absence, in these results, of a clear gel-to-liquid crystal or liquid crystal-to-gel phase transition is characteristic of archaeal diether lipid bilayers and a major difference from ...”.

Modified as suggested.

- There is a similar issue on lines 293 and 294 where it isn't clear if it is the “sharp phase transition” that is characteristic or “no sharp phase transition” that is characteristic. In this case, the intended meaning seems to be that it is the sharp phase transition that is characteristic of bacterial/eukaryal phospholipids. This could be made more clear with wording like “We observed no sharp phase transition in contrast to what would be expected for bacterial/eukaryal phospholipids.”

Modified as suggested.

- In line 305, the absence of a word after “therefore” may have affected the apparent meaning of this sentence. Should it read something like “reduces the frustration of the lipids and therefore enables formation of non-lamellar phases.”?

Thank you for noticing it. Indeed, the meaning was not that it reduces the formation as the absence of a word would lead to understand, but on the contrary that it increases the formation. We have modified it as suggested.

- The text between lines 454 and 458 has been modified slightly to better reflect the logic behind extending the findings here, on the model system, to a real system of archaeal lipids. I think that the wording is an improvement but the meaning could be made clearer by a slight change in emphasis. Perhaps something like “For the archaeal lipid analogs used here, we did not observe these large negative curvatures in the absence of squalene, even at the most extreme P and T conditions tested. This suggests that without apolar intercalants, the classical amphiphilic archaeal lipids mixtures modeled by our system might also lack the ability to modulate membrane bending under the physiological conditions of the archaeon.”

Modified as suggested.

I would like to again emphasize that, on balance, I find this to be an interesting, significant, and well executed contribution. I did find some additional minor issues with wording as noted below.

- **Minor suggestions and corrections:**

All minor suggestions have been taken into account.

Line 44: Suggest "... and are indispensable ..."

Line 52: Suggest removing "the" so that the sentence starts "Over time, ..."

Line 58: "... lipid .." should be plural.

Line 72: Should be "the ratio of..."

Line 133: For number sense to agree, it should be either "... lipid compositions in natural membranes are..." or "... lipid composition in natural membranes is ..."

Line 143: Should "of" be replaced by "or" so that it reads "T. barophilus or M. kandleri"?

Also, in this section, the usage seems to vary between T. barophilus and Thermococcus barophilus. Perhaps these names should be written out in full once and then in the short version consistently after that. There are other places in the manuscript where the usage seems to alternate between full and shortened nomenclature.

Line 169: Should be a space after "(DB)".

Line 214: The wording "...does not allow really to extrapolate at length..." is awkward. Maybe something more direct like "...does not provide any additional information about the distinction between these coexisting lamellar phases."

We changed the sentence as suggested.

The table in Figure 3 contains the label "N.D." in the entries for lamellar at 70°C and 85°C. I am guessing that this means "not detected" but I realized that this doesn't seem to be defined anywhere.

Line 277: There is an issue with wording here. It could be addressed by adding "for" before "which".

Line 287: "and" should be "an" so that it reads "...associated with an increase...".

Line 329: Should insert "in" after "increase".

Line 331: Should insert "than" after "extent".

Line 352: Suggest rewording to read "... allows permeability to reach a level that is...".

Line 387: Suggest rewording to "...but exponentially and with inverse proportionality to membrane thickness..".

Line 391: Should this read "...induces the lateral reorganization into domains corresponding to...".

Reviewer #3 (Remarks to the Author):

Overall, the manuscript has been considerably improved, although there are some typos and/or small grammatical errors that might be corrected.

The manuscript has been carefully edited to avoid these typos. Hopefully we have removed them all now.

I still have a few **minor comments**:

- Lines 66-67, "The recent discovery of monolayer, ether lipids in the membranes of the most hyperthermophilic bacteria gave further support to the view that tetra-ether lipids were the adaptation route to life at high temperatures." The authors do not provide a reference here, but they should. Ether-linked lipids in hyperthermophilic bacteria have been known for over 20 years now, although some recent papers just ignore that information. It would be good to provide appropriate original citations.

122°C as an upper temperature for *Methanopyrus kandlerii*. The explanation of the authors would be adequate in an ideal world. However, it is not because something is published that it is necessarily true and does not need validation. Life at 250°C was reported in one top-journal, never to be reproduced! 122°C may need validation as well.

We couldn't agree more with this reviewer. We have not been able to reproduce it ourselves in the lab yet. This has been an issue as well for the high pressure work published by Sharma and colleagues in *Science*, which has plagued all further and sound publications afterwards. In order to reflect this this sentence now reads "the alleged current record holder...".

- That membrane-bending proteins have an effect on membrane remodelling is undeniable. That this is linked to particular membrane phospholipid rearrangements is possible. The authors should recognize this. A clear example is that of membrane-remodelling proteins in Asgard archaea. The particular morphology of *Promethoarchaeum syntrophicum* with extensive membrane protrusions is most likely largely caused by membrane-remodelling ESCRT-like proteins.

We have added the following sentences to emphasize this point. " In particular, the role of membrane-bending proteins on membrane remodeling cannot be ignored in Archaea, as shown in Asgard archaea in which the particular morphology of *Promethoarchaeum syntrophicum* with extensive membrane protrusions is most likely caused by membrane-remodelling ESCRT-like proteins. Hence, if demonstrated, the ability of apolar polyisoprenoid will not be the only means by which Archaea might introduce high bending in their membranes, but might play a role in specific instances in which membrane proteins are not favorable."